# A Simple and Effective Reinforcement Learning Method for Text-to-Image Diffusion Fine-tuning

**Shashank Gupta**[*][‡]                                          *27392shashankgupta@gmail.com*
*University of Amsterdam, The Netherlands*

**Chaitanya Ahuja**                                                      *chahuja@meta.com*
*Meta*

**Tsung-Yu Lin**                                                      *tsungyulin@meta.com*
*Meta*

**Sreya Dutta Roy**                                                  *sreyaduttaroy@meta.com*
*Meta*

**Harrie Oosterhuis**                                              *harrie.oosterhuis@ru.nl*
*Radboud University, Nijmegen, The Netherlands*

**Maarten de Rijke**                                                      *m.derijke@uva.nl*
*University of Amsterdam, The Netherlands*

**Satya Narayan Shukla**[§]                                          *sns.iitkgp@gmail.com*
*Meta*

**Reviewed on OpenReview:** *https://openreview.net/forum?id=i8WJhKn455*

## Abstract

Reinforcement learning (RL)-based fine-tuning has emerged as a powerful approach for aligning diffusion models with black-box objectives. Proximal policy optimization (PPO) is a popular choice of method for policy optimization. While effective in terms of performance and sample complexity, PPO is highly sensitive to hyper-parameters and involves substantial computational overhead. REINFORCE, on the other hand, mitigates some implementation complexities such as high memory overhead and sensitive hyper-parameter tuning, but has suboptimal performance due to high variance and crucially sample inefficiency, which is the primary notion of efficiency we study in this work. While the variance of the REINFORCE can be reduced by sampling multiple actions per input prompt and using a baseline correction term, it still suffers from sample inefficiency. To address these challenges, we systematically analyze the sample efficiency-effectiveness trade-off between REINFORCE and PPO, and propose *leave-one-out PPO* (LOOP), a novel RL for diffusion fine-tuning method. LOOP combines variance reduction techniques from REINFORCE, such as sampling multiple actions per input prompt and a baseline correction term, with the robustness and sample efficiency of PPO via clipping and importance sampling. Our results demonstrate that LOOP effectively improves diffusion models on various black-box objectives, and achieves a better balance between sample efficiency and final performance.

---

[*]Corresponding author.
[‡]Work done during an internship at Meta AI.
[§]Corresponding author.

# 1 Introduction

Diffusion models have emerged as a powerful tool for generative modeling (Ho et al., 2020; Sohl-Dickstein et al., 2015), with a strong capacity to model complex data distributions from various modalities, like images (Rombach et al., 2022), text (Austin et al., 2021), natural molecules (Xu et al., 2023), and videos (Blattmann et al., 2023).

Diffusion models are typically pre-trained on a large-scale dataset, such that they can subsequently generate samples from the same data distribution. The training objective typically involves maximizing the data distribution likelihood. This pre-training stage helps generate high-quality samples from the model. However, some applications might require optimizing a custom reward function, for example, optimizing for generating aesthetically pleasing images (Xu et al., 2024), semantic alignment of image-text pairs based on human feedback (Schuhmann et al., 2022), or generating molecules with specific properties (Wang et al., 2024).

To optimize for such black-box objectives, RL-based fine-tuning has been successfully applied to diffusion models (Black et al., 2023; Fan et al., 2024; Gu et al., 2024; Li et al., 2024; Wallace et al., 2024). For RL-based fine-tuning, the reverse diffusion process is treated as a Markov decision process (MDP), wherein prompts are treated as part of the input state, the generated image at each time-step is mapped to an action, which receives a reward from a fixed reward model (environment in standard MDP), and finally the diffusion model is treated as a policy, which we optimize to maximize rewards. For optimization, typically PPO is applied (Black et al., 2023; Fan et al., 2024). In applications where getting a reward model is infeasible or undesirable, "RL-free" fine-tuning (typically offline) can also be applied (Wallace et al., 2024). For this work, we only focus on diffusion model fine-tuning using "online" RL methods, specifically policy-gradient style methods, like PPO (Schulman et al., 2017).

An advantage of PPO is that it removes the incentive for the new policy to deviate too much from the previous reference policy, via importance sampling and clipping (Schulman et al., 2017), which, as we discuss below, leads to superior sample efficiency over simpler policy-gradient methods such as REINFORCE (Williams, 1992). However, PPO comes with significant implementation overhead: in practice, RL fine-tuning for diffusion models via PPO requires concurrently loading three models in memory: (i) The **reference policy:** The base policy, usually initialized with the pre-trained diffusion model. (ii) The **current policy:** The RL fine-tuned policy, also initialized with the pre-trained diffusion model. (iii) The **reward model:** Typically a large vision-language model trained via supervised fine-tuning (Lee et al., 2023), which assigns a scalar reward to the final generated image. In addition, PPO is known to be sensitive to hyper-parameters (Engstrom et al., 2019; Huang et al., 2024; Zheng et al., 2023). We note these costs as practical motivation for seeking simpler alternatives, though the primary axis of comparison in this work is *sample efficiency*, not computational cost.

Simpler approaches, such as REINFORCE (Williams, 1992), avoid such additional implementation complexities by requiring neither a separate reference policy in memory nor sensitive hyperparameter tuning. However, this implementation simplicity comes at a cost: in practice, REINFORCE is *less* sample efficient than PPO, and suffers from high variance and instability (Gupta, 2025; Gupta et al., 2024b). A variant of REINFORCE, reinforce leave-one-out (RLOO) (Ahmadian et al., 2024; Kool et al., 2019), samples multiple sequences per input prompt and applies a baseline correction term to reduce variance; however, it remains sample inefficient because it does not permit trajectory reuse across policy updates, unlike importance-sampling-based methods.

This raises a fundamental question about the **efficiency-effectiveness** trade-off in RL-based diffusion fine-tuning. In this work, we systematically explore the trade-off between sample efficiency, defined as requiring fewer training prompts to achieve good performance, and effectiveness, measured by stable training and final reward. We compare a simple REINFORCE approach with the standard PPO framework, demonstrating that while REINFORCE greatly reduces implementation complexity, it comes at the cost of reduced performance and sample complexity.

Motivated by this finding, we propose a novel RL method for diffusion fine-tuning, LOOP, which combines the best of both worlds. To reduce the variance during policy optimization, LOOP uses multiple actions (diffusion trajectories) and a REINFORCE-style baseline correction term per input prompt. To maintain the stability, robustness, and sample complexity of PPO, LOOP uses clipping and importance sampling.

We clarify an important distinction regarding the notion of efficiency in this work. When discussing the efficiency-effectiveness trade-off, we primarily refer to sample efficiency, defined as an algorithm's ability to achieve better performance with the same number of input prompts during training. LOOP exhibits superior sample efficiency compared to PPO. We emphasize this notion because it directly translates into better performance for a fixed training dataset size, which is often the dominant constraint in practice when optimizing diffusion models with computationally expensive reward models. For a fixed number of training prompts, LOOP attains higher reward values by sampling multiple trajectories per prompt and employing a leave-one-out baseline correction term.

However, we note that while LOOP demonstrates superior sample efficiency by requiring fewer training prompts to achieve a given performance level, it requires $K$ diffusion sampling passes per prompt, leading to an extra $O(K)$ computational overhead relative to PPO. Future work could explore adaptive sampling strategies, asynchronous generation pipelines (Han et al., 2025), or distributed trajectory sampling (Bartoldson et al., 2025) to mitigate this computational cost while preserving sample efficiency gains. We report the total GPU runtime and memory usage in Appendix E. We leave the study and improvement of the computational efficiency of LOOP as part of future work.

Our approach is conceptually similar to the recently proposed GRPO method for RL fine-tuning of LLMs (Shao et al., 2024). The key technical differences are: (i) LOOP does not apply standard-deviation normalization in the advantage calculation. Recent work on LLM fine-tuning suggests that removing this normalization term can improve performance (Liu et al., 2025). (ii) Following this recent work, LOOP omits the KL penalty term. Prior studies indicate that explicit KL regularization has minimal practical effect on performance (Black et al., 2023), and recent theoretical work shows that on-policy RL methods implicitly maintain KL proximity to the base policy even without explicit regularization (Shenfeld et al., 2025). (iii) In the diffusion setting, the reverse process has a fixed sequence length across all generations, making sequence-length normalization unnecessary. We provide an empirical comparison on GRPO vs. LOOP in Appendix B.

For the primary evaluation benchmark, we choose the text-to-image compositionality benchmark T2I-CompBench (Huang et al., 2023). Text-to-image models often fail to satisfy an essential reasoning ability of attribute binding, i.e., the generated image often fails to *bind* certain *attributes* specified in the instruction prompt (Fu & Cheng, 2024; Huang et al., 2023; Ramesh et al., 2022). As illustrated in Figure 1, LOOP outperforms previous diffusion methods on attribute binding. As attribute binding is a key skill necessary for real-world applications, we choose the T2I-CompBench benchmark alongside two other common tasks: aesthetic image generation and image-text semantic alignment (Black et al., 2023).

To summarize, our main contributions are as follows:

- **PPO vs. REINFORCE efficiency-effectiveness trade-off.** We systematically study how design elements like clipping, reference policy, value function in PPO compare to a simple REINFORCE method, highlighting the efficiency-effectiveness trade-off in diffusion fine-tuning. To the best of our knowledge, we are the first ones to present such a systematic study, highlighting the trade-offs in diffusion fine-tuning.

- **Introducing LOOP.** We propose LOOP, a novel RL for diffusion fine-tuning method combining the best of REINFORCE and PPO. LOOP uses multiple diffusion trajectories and a REINFORCE baseline correction term for variance reduction, as well as clipping and importance sampling from PPO for robustness and sample efficiency.

- **Empirical validation.** To validate our claims empirically, we conduct experiments on the T2I-CompBench benchmark image compositionality benchmark. The benchmark evaluates the attribute binding capabilities of the text-to-image generative models and shows that LOOP succeeds where previous text-to-image generative models often fail. We also evaluate LOOP on two common objectives from the literature on RL for diffusion: image aesthetic and text-image semantic alignment (Black et al., 2023).

The remainder of the paper is organized as follows. In the next section, we provide the necessary background and discuss related work. Section 3 revisits the efficiency–effectiveness trade-off between REINFORCE and

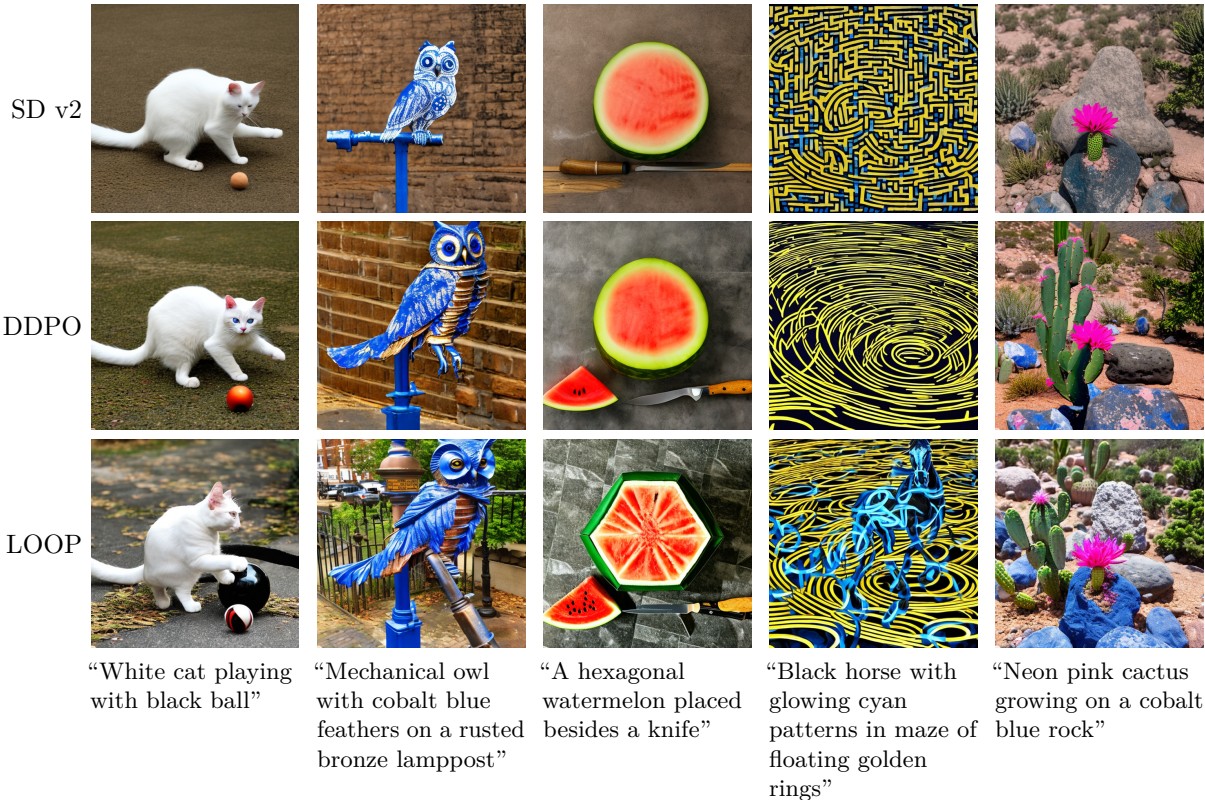

SD v2

DDPO

LOOP

"White cat playing with black ball"    "Mechanical owl with cobalt blue feathers on a rusted bronze lamppost"    "A hexagonal watermelon placed besides a knife"    "Black horse with glowing cyan patterns in maze of floating golden rings"    "Neon pink cactus growing on a cobalt blue rock"

Figure 1: **LOOP improves attribute binding**. Qualitative examples presented from images generated via Stable Diffusion (SD) 2.0 (first row), DDPO (Black et al., 2023) (second row), and LOOP $k = 4$ (third row). In the first prompt, SD and DDPO both fail to bind the *color black* with the *ball* in the image, whereas LOOP binds the color black to the ball. In the second example, SD and DDPO fail to generate *rusted bronze color* lamppost, whereas LOOP manages to do that. In the third image, SD and DDPO fail to bind the *shape hexagon* to the watermelon, whereas LOOP manages so. In the fourth example, SD and DDPO fail to generate the *black horse* with flowing cyan patterns, whereas LOOP generates the horse with the correct color attribute. Finally, in the last image, SD and DDPO fail to bind *cobalt blue* color to the rock, whereas LOOP binds that successfully.

PPO. Section 4 introduces our proposed method, Leave-One-Out PPO (LOOP) for diffusion fine-tuning. Section 5 describes the experimental setup, Section 6 details our hyperparameters, and Sections 7 and 8 present our results and qualitative examples. Finally, Section 9 concludes the paper.

## 2 Background and Related Work

### 2.1 Diffusion Models

We focus on denoising diffusion probabilistic models (DDPM) as the base model for text-to-image generative modeling (Ho et al., 2020; Sohl-Dickstein et al., 2015). Briefly, given a conditioning context variable $\mathbf{c}$ (a text prompt in our case), and the data sample $\mathbf{x}_0$, DDPM models $p(\mathbf{x}_0 \mid \mathbf{c})$ via a Markov chain of length $T$, with the following dynamics:

$$p_\theta(\mathbf{x}_{0:T} \mid \mathbf{c}) = p(\mathbf{x}_T \mid \mathbf{c}) \prod_{t=1}^{T} p_\theta(\mathbf{x}_{t-1} \mid \mathbf{x}_t, \mathbf{c}). \tag{1}$$

Image generation in a diffusion model is achieved via the following ancestral sampling scheme, which is a reverse diffusion process:

$$\mathbf{x}_T \sim \mathcal{N}(\mathbf{0}, \mathbf{I}), \quad \mathbf{x}_t \sim N\left(\mathbf{x}_t \mid \mu_\theta(\mathbf{x}_t, \mathbf{c}, t), \sigma_\theta^2 I\right), \forall t \in [0, T-1], \tag{2}$$

where the distribution at time-step $t$ is assumed to be a multivariate normal distribution with the predicted mean $\mu_\theta(\mathbf{x}_t, \mathbf{c}, t)$, and a constant variance.

## 2.2 Proximal Policy Optimization (PPO) for RL

PPO was introduced for optimizing a policy with the objective of maximizing the overall reward in the RL setup. PPO removes the incentive for the current policy $\pi_t$ to diverge from the previous policy $\pi_{t-1}$ outside the range $[1 - \epsilon, 1 + \epsilon]$, where $\epsilon$ is a hyper-parameter. As long as the subsequent policies are closer to each other in the action space, the monotonic policy improvement bound guarantees a monotonic improvement in the policy's performance as the optimization progresses. This property justifies the clipping term in the mathematical formulation of the PPO objective function (Achiam et al., 2017; Queeney et al., 2021; Schulman, 2015). Formally, the PPO objective function is:

$$J(\theta) = \mathbb{E}\Big[\min\Big(r_t(\theta)\,\hat{\mathbf{A}}_t,\, \text{clip}(r_t(\theta), 1 - \epsilon, 1 + \epsilon)\hat{\mathbf{A}}_t\Big)\Big], \tag{3}$$

where $r_t(\theta) = \frac{\pi_t(a|c)}{\pi_{t-1}(a|c)}$ is the importance sampling ratio between the current policy $\pi_t(a \mid c)$ and the previous reference policy $\pi_{t-1}(a \mid c)$, $\hat{A}_t$ is the advantage function (Sutton & Barto, 2018), and the clip operator restricts the importance sampling ratio in the range $[1 - \epsilon, 1 + \epsilon]$.

## 2.3 RL for Text-to-Image Diffusion Models

The diffusion process can be viewed as an MDP $(\mathcal{S}, \mathcal{A}, \mathcal{P}, \mathcal{R}, \rho_0)$, where $\mathcal{S}$ is the state space, $\mathcal{A}$ is the action space, $\mathcal{P}$ is the state transition kernel, $\mathcal{R}$ is the reward function, and $\rho_0$ is the distribution of initial state $\mathbf{s_0}$. In the context of text-to-image diffusion models, the MDP is defined as:

$$\mathbf{s_t} = (\mathbf{c}, t, \mathbf{x_t}), \quad \pi_\theta(\mathbf{a_t} \mid \mathbf{s_t}) = p_\theta(\mathbf{x_{t-1}} \mid \mathbf{x_t}, \mathbf{c}), \quad \mathcal{P}(\mathbf{s_{t+1}} \mid \mathbf{s_t}, \mathbf{a_t}) = \delta(\mathbf{c}, \mathbf{a_t}), \quad \mathbf{a_t} = \mathbf{x_{t-1}},$$

$$\rho_0(\mathbf{s_0}) = \big(p(\mathbf{c}), \delta_T, \mathcal{N}(0, \mathbf{I})\big), \quad \mathcal{R}(\mathbf{s_t}, \mathbf{a_t}) = \begin{cases} r(\mathbf{x_0}, \mathbf{c}) & \text{if } t = 0, \\ 0 & \text{otherwise.} \end{cases} \tag{4}$$

The input state $\mathbf{s_t}$ is defined in terms of the context $\mathbf{c}$ (prompt features), and the sampled image at the given time-step $t$: $\mathbf{x_t}$. The policy $\pi_\theta$ is the diffusion model itself. The state transition kernel is a dirac delta function $\delta$ with the current sampled action $\mathbf{x}_t$ as the input. The reward is assigned only at the last step in the reverse diffusion process, when the final image is generated. The initial state $\rho_0$ corresponds to the last state in the forward diffusion process: $\mathbf{x}_T$.

## 2.4 PPO for Diffusion Fine-tuning

The objective function of RL fine-tuning for a diffusion policy $\pi_\theta$ can be defined as follows:

$$J_\theta(\pi) = \mathbb{E}_{\tau \sim p(\tau|\pi_\theta)}\left[\sum_{t=0}^{T} \mathcal{R}(\mathbf{s_t}, \mathbf{a}_t)\right] = \mathbb{E}_{\tau \sim p(\tau|\pi_\theta)}\left[r(\mathbf{x_0}, \mathbf{c})\right], \tag{5}$$

where the trajectory $\tau = \{\mathbf{x}_T, \mathbf{x}_{T-1}, \ldots, \mathbf{x}_0\}$ refers to the reverse diffusion process (Eq. 1), and the total reward of the trajectory is the reward of the final generated image $\mathbf{x}_0$ (Eq. 4). We ignore the KL-regularized version of the equation, which is commonly applied in the RLHF for LLM literature (Rafailov et al., 2023; Zeng et al., 2024; Zhong et al., 2024), and proposed by Fan et al. (2024) in the context of RL for diffusion models. As shown by Black et al. (2023), adding the KL-regularization term makes no empirical difference in terms of the final performance. The PPO objective is given as:

$$J_\theta^{\text{PPO}}(\pi) = \mathbb{E}\left[\sum_{t=0}^{T} \text{clip}\left(\frac{\pi_\theta(\mathbf{x}_{t-1}|\mathbf{x}_t, \mathbf{c})}{\pi_{\text{old}}(\mathbf{x}_{t-1}|\mathbf{x}_t, \mathbf{c})}, 1 - \epsilon, 1 + \epsilon\right) r(\mathbf{x_0}, \mathbf{c})\right],$$

where the clipping operation removes the incentive for the new policy $\pi_\theta$ to differ from the previous round policy $\pi_{\text{old}}$ (Black et al., 2023; Schulman et al., 2017).

## 2.5 Relationship to Offline Preference Based Methods

Although LOOP operates in an online reinforcement learning setting, similar to PPO, recent work has explored adapting direct preference optimization (DPO) (Rafailov et al., 2023) and related offline methods to diffusion models.

**DPO-based diffusion alignment**: Several methods have extended the DPO framework to diffusion models. Diffusion DPO (Rafailov et al., 2023; Wallace et al., 2024) applies the DPO objective to diffusion models by treating denoising as a sequential decision making process and by training on preference pairs without explicit reward queries. Diffusion RPO (Gu et al., 2024) introduces pairwise preference optimization at each denoising timesteps, as opposed to pairwise preference at the trajectory level. Other work has explored noise conditioned preference learning (Gambashidze et al., 2024) as well as applications to diffusion based policy learning (Kang et al., 2023).

We do not include direct empirical comparisons with DPO style methods because these approaches learn from pre-collected preference datasets rather than from online reward queries and thus represent a complementary paradigm for aligning generative models with human preferences. DPO-based methods rely on pre-collected preference datasets and are therefore constrained by the coverage of their training data, whereas online methods such as LOOP query rewards during training and can explore beyond the initial policy distribution. In settings where large and high quality preference datasets are available, DPO style methods may offer competitive performance with fewer online reward queries. Conversely, when online exploration is feasible and preference data is limited, online methods may be more appropriate. Future work could investigate systematic comparisons between online and offline methods under controlled conditions, as well as hybrid approaches that combine both paradigms.

## 3 REINFORCE vs. PPO: A Sample Efficiency-Effectiveness Trade-Off

In this section, we explore the sample efficiency-effectiveness trade-off between two prominent reinforcement learning methods for diffusion fine-tuning: REINFORCE and PPO. Understanding this trade-off is crucial for selecting the appropriate algorithm given constraints on training data or reward-query budgets.

In the context of text-to-image diffusion models, we aim to optimize the policy $\pi$ to maximize the expected reward $\mathcal{R}(\mathbf{x}_{0:T}, \mathbf{c}) = r(\mathbf{x}_0, \mathbf{c})$. Our objective function is defined as:

$$J_\theta(\pi) = \mathbb{E}_{\mathbf{c} \sim p(\mathbf{C}), \mathbf{x}_{0:T} \sim p_\theta(\mathbf{x}_{0:T} | \mathbf{c})} \left[ r(\mathbf{x}_0, \mathbf{c}) \right]. \tag{6}$$

**REINFORCE for gradient calculation.** For optimizing this objective, the REINFORCE policy gradient (also known as score function (SF)) (Williams, 1992) provides the following gradient estimate:

$$\begin{aligned}
&\nabla_\theta J_\theta^{\mathrm{SF}}(\pi) \\
&= \mathbb{E}_{\mathbf{x}_{0:T}} \left[ \nabla_\theta \log \left( \prod_{t=1}^T p_\theta(\mathbf{x}_{t-1} \mid \mathbf{x}_t, \mathbf{c}) \right) r(\mathbf{x}_0, \mathbf{c}) \right] \\
&= \mathbb{E}_{\mathbf{x}_{0:T}} \left[ \sum_{t=0}^T \nabla_\theta \log p_\theta(\mathbf{x}_{t-1} \mid \mathbf{x}_t, \mathbf{c}) r(\mathbf{x}_0, \mathbf{c}) \right],
\end{aligned} \tag{7}$$

where the second step follows from the reverse diffusion policy decomposition (Eq. 1).

In practice, a batch of trajectories is sampled from the reverse diffusion distribution, i.e., $\mathbf{x}_{0:T} \sim p_\theta(\mathbf{x}_{0:T})$, and a Monte-Carlo estimate of the REINFORCE policy gradient (Eq. 7) is calculated for the model update.

**REINFORCE with baseline correction.** To reduce variance of the REINFORCE estimator, a common trick is to subtract a constant baseline correction term from the reward function (Greensmith et al., 2004; Gupta et al., 2024b; Mohamed et al., 2020):

$$\nabla_\theta J_\theta^{\mathrm{SFB}}(\pi) = \mathbb{E} \left[ \sum_{t=0}^T \nabla_\theta \log p_\theta(\mathbf{x}_{t-1} \mid \mathbf{x}_t, \mathbf{c})(r(\mathbf{x}_0, \mathbf{c}) - b_t) \right]. \tag{8}$$

**REINFORCE Leave-one-out (RLOO).** To further reduce the variance of the REINFORCE estimator, RLOO samples $K$ diffusion trajectories per prompt ($\{\mathbf{x}_{0:T}^k\} \sim \pi(. \mid \mathbf{c})$), for a better Monte-Carlo estimate of the expectation (Ahmadian et al., 2024; Kool et al., 2019). The RLOO estimator is:

$$\nabla_\theta J_\theta^{\mathrm{RLOO}}(\pi) = \mathbb{E}\left[K^{-1} \sum_{k=0}^{K} \sum_{t=0}^{T} \nabla_\theta \log p_\theta(\mathbf{x}_{t-1}^k \mid \mathbf{x}_t^k, \mathbf{c})\big(r(\mathbf{x}_0^k, \mathbf{c}) - b_t\big)\right]. \tag{9}$$

However, REINFORCE-based estimators have a significant disadvantage: they do not allow sample reuse (i.e., reusing trajectories collected from previous policies) due to a distribution shift between policy gradient updates during training. Sampled trajectories can only be used once, prohibiting mini-batch updates. This makes it *sample inefficient.*

To allow for sample reuse, the importance sampling (IS) trick can be applied (Owen, 2013; Schulman, 2015):

$$J_\theta^{\mathrm{IS}}(\pi) = \mathbb{E}_{\mathbf{c}_t \sim p(\mathbf{C}), \mathbf{a}_t \sim \pi_{\mathrm{old}}(\mathbf{a}_t \mid \mathbf{c}_t)}\left[\frac{\pi_\theta(\mathbf{a}_t \mid \mathbf{c}_t)}{\pi_{\mathrm{old}}(\mathbf{a}_t \mid \mathbf{c}_t)} \mathcal{R}_t\right], \tag{10}$$

where $\pi_\theta$ is the *current* policy to be optimized, and $\pi_{\mathrm{old}}$ is the policy from the previous update round. With the IS trick, we can sample trajectories from the current policy in a batch, store it in a temporary buffer, and re-use them to apply mini-batch optimization (Schulman et al., 2017).

**Motivation for PPO.** With the IS trick, the samples from the old policy can be used to estimate the policy gradient under the current policy $\pi_\theta$ (Eq. 7) in a statistically unbiased fashion (Owen, 2013), i.e., in expectation the IS and REINFORCE gradients are equivalent (Eq. 10, Eq. 7). Thus, potentially, we can improve the sample efficiency of REINFORCE gradient estimation with IS.

While unbiased, the IS estimator can exhibit high variance (Owen, 2013). This high variance may lead to unstable training dynamics. Additionally, significant divergence between the current policy $\pi_\theta$ and the previous policy $\pi_{\mathrm{old}}$ can result in the updated diffusion policy performing worse than the previous one (Achiam et al., 2017; Schulman, 2015). Next, we will prove this formally. We note that this result has previously been established by (Achiam et al., 2017) for the more general RL setting. In this work, we extend this finding to the context of diffusion model fine-tuning.

A key component of the proof relies on the distribution of states under the current policy, i.e., $d^\pi(\mathbf{s})$. In the case of diffusion models, the state transition kernel $P(\mathbf{s_{t+1}} \mid \mathbf{s_t}, \mathbf{a_t})$ is deterministic, because the next state consists of the action sampled from the previous state (Eq. 4), i.e., $P(\mathbf{s_{t+1}} \mid \mathbf{s_t}, \mathbf{a_t}) = 1$. While the state transition kernel is deterministic, the distribution of states is stochastic, given that it depends on the action at time $t$, which is sampled from the policy (Eq. 4). We define the state distribution as follows.

**Definition 1.** *Given the distribution over contexts $\mathbf{c} \sim p(\mathbf{C})$, the (deterministic) distribution over time $t = \delta(t)$, and the diffusion policy $\pi$, the state distribution at time $t$ is:*

$$p(\mathbf{s}_t \mid \pi) = p(\mathbf{c})\delta(t)\int_{\mathbf{x}_{t+1}} \pi(\mathbf{x}_t \mid \mathbf{x}_{t+1}, \mathbf{c}, t)\pi(\mathbf{x}_{t+1} \mid \mathbf{c}, t) \, \mathbf{dx}_{t+1}.$$

Subsequently, the normalized discounted state visitation distribution can be defined as:

$$d^\pi(\mathbf{s}) = (1 - \gamma) \sum_{t=0}^{\infty} \gamma^t p(\mathbf{s}_t = \mathbf{s} \mid \pi). \tag{11}$$

The advantage function is defined as: $A^{\pi_k}(\mathbf{s}, \mathbf{a}) = Q^{\pi_k}(\mathbf{s}, \mathbf{a}) - V^{\pi_k}(\mathbf{s})$ (Sutton & Barto, 2018). Given this, the monotonic policy improvement bound can be derived:

**Theorem 3.1** (Achiam et al., 2017). *Consider a current policy $\pi_k$. Let $C^{\pi, \pi_k} = \max_{\mathbf{s} \in S} \left|\mathbb{E}_{\mathbf{a} \sim \pi(\cdot \mid \mathbf{s})}\left[A^{\pi_k}(\mathbf{s}, \mathbf{a})\right]\right|$, and $\mathrm{TV}(\pi(\cdot \mid \mathbf{s}), \pi_k(\cdot \mid \mathbf{s}))$ represent the total variation distance between the policies $\pi(\cdot \mid \mathbf{s})$ and $\pi_k(\cdot \mid \mathbf{s})$, and $\mathbf{s}$ be the current state. For any future policy $\pi$, we have:*

$$J(\pi) - J(\pi_k) \geq \frac{1}{1-\gamma}\mathbb{E}_{(\mathbf{s}, \mathbf{a}) \sim d^{\pi_k}}\left[\frac{\pi(\mathbf{a} \mid \mathbf{s})}{\pi_k(\mathbf{a} \mid \mathbf{s})}A^{\pi_k}(\mathbf{s}, \mathbf{a})\right] - \frac{2\gamma C^{\pi, \pi_k}}{(1-\gamma)^2}\mathbb{E}_{\mathbf{s} \sim d^{\pi_k}}\left[\mathrm{TV}(\pi(\cdot \mid \mathbf{s}), \pi_k(\cdot \mid \mathbf{s}))\right].$$

A direct consequence of this theorem is that when optimizing a policy with the IS objective (Eq. 10), to guarantee that the new policy will improve upon the previous policy, the policies should not diverge too much. Therefore, we need to apply a constraint on the current policy. This can be achieved by applying the clipping operator in the PPO objective (Eq. 3) (Achiam et al., 2017; Gupta et al., 2024a;c; Queeney et al., 2021; Schulman et al., 2017). We provide an empirical comparison of PPO with and without clipping in Appendix D, demonstrating the practical importance of the clipping operator for stable and effective diffusion fine-tuning.

This gives rise to a *sample efficiency-effectiveness trade-off* between REINFORCE and PPO. REINFORCE offers greater implementation simplicity, requiring fewer hyperparameters and less memory overhead, but at the cost of lower sample efficiency and suboptimal final performance. PPO, despite its implementation overhead, achieves superior sample efficiency and reliable policy improvements through importance sampling and clipping.

We note that a similar trade-off analysis was performed in the context of RL fine-tuning for large language models (LLM) (Ahmadian et al., 2024). However, their analysis was limited to an empirical study, whereas we present a theoretical analysis in addition to the empirical analysis. To the best of our knowledge, we are the first to conduct such a study for diffusion methods.

## 4 Method: Leave-One-Out PPO (LOOP) for Diffusion Fine-tuning

We demonstrated the importance of PPO in enhancing sample efficiency and achieving stable improvements during training for diffusion fine-tuning. Additionally, we showcased the RLOO method's effectiveness in reducing the variance of the REINFORCE method.

In this section, we introduce our proposed method, **LOOP**, a novel RL for diffusion fine-tuning method combining the best of both worlds. While PPO achieves superior sample efficiency over REINFORCE, its objective is still estimated from a single trajectory per prompt, which can result in high-variance gradient estimates. We exploit this remaining source of variance as our starting point for improving upon PPO.

The expectation in the PPO loss (Eq. 3) is typically estimated by sampling a single trajectory from the policy in the previous iteration $\pi_{old}$: $\mathbf{x}_{0:T} \sim \pi_{old}$ for a given prompt $c$:

$$\sum_{t=0}^{T} \text{clip}\left( \frac{\pi_\theta(\mathbf{x}_{t-1}|\mathbf{x}_t, \mathbf{c})}{\pi_{\text{old}}(\mathbf{x}_{t-1}|\mathbf{x}_t, \mathbf{c})}, 1 - \epsilon, 1 + \epsilon \right) r(\mathbf{x}_0, \mathbf{c}). \tag{12}$$

Even though the single sample estimate is an unbiased Monte-Carlo approximation of the expectation, it can have potentially high-variance (Owen, 2013). Additionally, the IS term ($\frac{\pi_\theta(\mathbf{x}_{t-1}|\mathbf{x}_t,\mathbf{c})}{\pi_{\text{old}}(\mathbf{x}_{t-1}|\mathbf{x}_t,\mathbf{c})}$) can also contribute to high-variance of the PPO objective (Swaminathan & Joachims, 2015; Xie et al., 2023). Both factors combined, can lead to high-variance, and unstable training of the PPO.

Taking inspiration from RLOO (Eq. 9), we sample $K$ independent trajectories from the previous policy for a given prompt $c$, and apply a baseline correction term from each trajectory's reward, to reduce the variance of the estimator:

$$\hat{J}_\theta^{\text{LOOP}}(\pi) = \frac{1}{K} \sum_{i=1}^{K} \left[ \sum_{t=0}^{T} \text{clip}\left( \frac{\pi_\theta(\mathbf{x}_{t-1}^i|\mathbf{x}_t^i, c)}{\pi_{\text{old}}(\mathbf{x}_{t-1}^i|\mathbf{x}_t^i, c)}, 1 - \epsilon, 1 + \epsilon \right) \cdot \left( r(\mathbf{x}_0^i, \mathbf{c}) - b^i \right) \right], \tag{13}$$

where $\mathbf{x}_{0:T}^i \sim \pi_{old}, \forall i \in [1, K]$. The baseline correction term $b^i$ reduces the variance of the gradient estimate, while being unbiased in expectation (Gupta et al., 2023; 2024b; Mohamed et al., 2020). A simple choice of baseline correction can be the average reward across the $K$ trajectories. However, it results in a biased estimator (Kool et al., 2019). Therefore, we choose the leave-one-out average baseline, with average taken across all samples in the trajectory, except the current sample $i$, i.e.,

$$b^i = \frac{1}{k-1} \sum_{j \neq i} r(\mathbf{x}_0^j). \tag{14}$$

We note that the baseline DDPO (PPO) method for diffusion fine-tuning (Black et al., 2023) employs a different variance-reduction strategy. Specifically, DDPO uses a running-mean baseline, where the baseline term is computed as a running average of rewards for a given prompt across optimization steps. This running mean is subtracted from the reward to reduce gradient variance. In contrast, LOOP adopts a leave-one-out baseline computed across the $K$ independently sampled trajectories within the same optimization step. Unlike the running-mean baseline, which aggregates rewards across iterations and may introduce bias due to policy drift, the leave-one-out baseline remains unbiased within each batch of sampled trajectories and provides stronger variance reduction by leveraging multiple contemporaneous samples per prompt. We provide a direct empirical comparison between the running-mean baseline used in DDPO and the leave-one-out baseline used in LOOP in Appendix C.

Originally RLOO sampling and baseline corrections were proposed in the context of REINFORCE, with a focus on on-policy optimization (Ahmadian et al., 2024; Kool et al., 2019), whereas we are applying these in the off-policy step of PPO. We call this method *leave-one-out PPO* (LOOP).

Our approach is conceptually similar to the recently popular GRPO method for RL fine-tuning of LLMs (Shao et al., 2024). Although our work was developed independently before GRPO gained widespread recognition, we do not include a head-to-head comparison.

Technically, the distinction lies in following aspects: (i) unlike GRPO, our formulation does not apply standard-deviation normalization in the denominator, as this has been shown to potentially harm performance in recent LLM fine-tuning via RL studies (Liu et al., 2025), (ii) similar to GRPO, we omit a KL penalty term, since our empirical experiments showed that it has little practical benefit. Furthermore, a recent study showed that on-policy RL implicitly constrains the updated policy to remain close to the base policy under a KL divergence measure, even without an explicit KL penalty term (Shenfeld et al., 2025), and (iii) we ignore the generation-length normalization term. In the diffusion setting, this simplification is further justified by the fact that the sequence length of the reverse diffusion process is fixed across generations, rendering length normalization unnecessary. We provide an empirical comparison on GRPO vs. LOOP in Appendix B. We further evaluate whether removing the KL term from the objective reduces diversity in the generated images by examining the reward–diversity trade-off; qualitative results are presented in Appendix F.

Provenly, LOOP has lower variance than PPO:

**Proposition 4.1.** *The LOOP estimator $\hat{J}_\theta^{\mathrm{LOOP}}(\pi)$ (Eq. 13) has lower variance than the PPO estimator $\hat{J}_\theta^{\mathrm{PPO}}(\pi)$ (Eq. 12):*

$$\mathrm{Var}\left[\hat{J}_\theta^{\mathrm{LOOP}}(\pi)\right] < \mathrm{Var}\left[\hat{J}_\theta^{\mathrm{PPO}}(\pi)\right]. \tag{15}$$

*Proof.* Since the sampled trajectories are independent:

$$\mathrm{Var}\left[\hat{J}_\theta^{\mathrm{LOOP}}(\pi)\right] = \frac{1}{K^2}\mathrm{Var}\left[\hat{J}_\theta^{\mathrm{PPO}}(\pi)\right] < \mathrm{Var}\left[\hat{J}_\theta^{\mathrm{PPO}}(\pi)\right]. \ \square$$

## 5 Experimental Setup

**Benchmark.** Text-to-image diffusion and language models often fail to satisfy an essential reasoning skill of attribute binding. Attribute binding reasoning capability refers to the ability of a model to generate images with attributes such as color, shape, texture, spatial alignment, (and others) specified in the input prompt. In other words, generated images often fail to *bind* certain *attributes* specified in the instruction prompt (Fu & Cheng, 2024; Huang et al., 2023; Ramesh et al., 2022). Since attribute binding seems to be a basic requirement for useful real-world applications, we choose the T2I-CompBench benchmark (Huang et al., 2023), which contains multiple attribute binding/image compositionality tasks, and its corresponding reward metric to benchmark text-to-image generative models. We also select two common tasks from prior RL for diffusion work: improving aesthetic quality of generation, and image-text semantic alignment (Black et al., 2023; Fan et al., 2024). To summarize, we choose the following tasks for the RL optimization: (i) Color, (ii) Shape, (iii) Texture, (iv) 2D Spatial, (v) Numeracy, (vi) Aesthetic, (vii) Image-text Alignment. For all tasks, the prompts are split into training/validation prompts. We report the average reward on both training and validation split.

**Model.** As the base diffusion model, we use Stable diffusion V2 (Rombach et al., 2022), which is a latent diffusion model. For optimization, we fully update the UNet model, with a learning rate of $1e^{-5}$. We also tried LORA fine-tuning (Hu et al., 2021), but the results were not satisfactory, so we update the entire model instead.

## 6 Hyperparameter and Implementation Details

For REINFORCE (including REINFORCE with baseline correction term), PPO, and LOOP the number of denoising steps ($T$) is set to 50. The diffusion guidance weight is set to 5.0. For optimization, we use AdamW (Loshchilov & Hutter, 2017) with a learning rate of $1e^{-5}$, and the weight decay of $1e-4$, with other parameters kept at the default value. We clip the gradient norm to 1.0. We train all models using 8 A100 GPUs with a batch size of 4 per GPU. The clipping parameter $\epsilon$ for PPO, and LOOP is set to $1e^{-4}$.

For all experiments, we use a DDIM sampler with 50 inference steps, set the noise parameter to $\eta = 1.0$, and apply classifier-free guidance with a guidance scale of 5.0.

For details of the implementation and a complete pseudo-code of the LOOP algorithm, we refer the reader to Appendix A.

## 7 Results and Discussion

### 7.1 REINFORCE vs. PPO Efficiency-Effectiveness Trade-off

We present our empirical results on the efficiency-effectiveness trade-off between REINFORCE and PPO. Our evaluation compares the following methods: the **REINFORCE** policy gradient for diffusion fine-tuning (Eq. 7); the REINFORCE policy gradient with a baseline correction term (**REINFORCE w/ BC**), detailed in Eq. 8, where the baseline term is the average reward for the given prompt (Black et al., 2023), and the **PPO** objective for diffusion fine-tuning, which incorporates importance sampling and clipping, as outlined in Eq. 3. This PPO objective is equivalent to the DDPO objective in the original RL for diffusion method (Black et al., 2023).

Figure 2 shows the training reward over epochs for the attributes: Color, Shape, and Texture from the T2I-CompBench benchmark, and training reward from optimizing the aesthetic model. Results are averaged over 3 runs. It is clear that REINFORCE policy gradient is not effective in terms of performance, as compared to other variants. Adding a baseline correction term indeed improves the training performance, validating the effectiveness of baseline in terms of training performance, possibly because of reduced variance. PPO achieves the highest training reward, validating the effectiveness of importance sampling and clipping for diffusion fine-tuning.

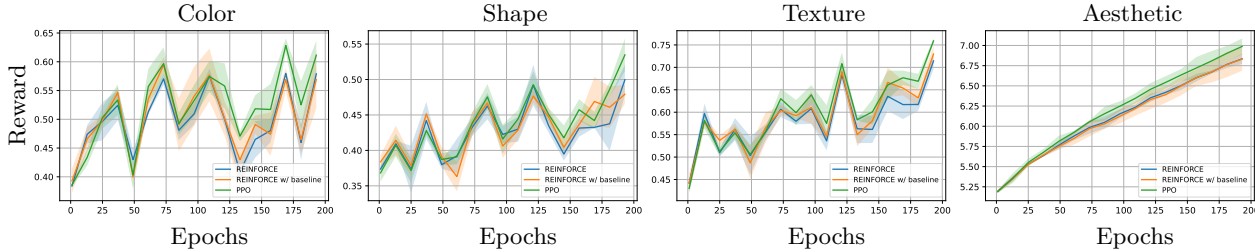

Figure 2: Evaluating REINFORCE vs. PPO trade-off by comparing: REINFORCE (Eq. 7), REINFORCE with baseline correction term (Eq. 8), and PPO (Eq. 3). We evaluate on the T2I-CompBench benchmark over three image attributes: Color, Shape, and Texture. We also compare on the aesthetic task. Y-axis corresponds to the training reward, x-axis corresponds to the training epoch. Results are averaged over three independent runs; shaded regions indicate $\pm 1$ standard deviation around the mean, computed over three independent runs.

We also evaluate the performance on a separate validation set. For each validation prompt, we generate 10 independent images from the diffusion policy, and average the reward, finally averaging over all evaluation prompts. The validation results are reported in Table 1. The results are consistent with the pattern observed with the training rewards, i.e., REINFORCE with baseline provides a better performance than plain REINFORCE, suggesting that baseline correction indeed helps with the final performance. Nevertheless, PPO (DDPO) still performs better than REINFORCE.

Table 1: Comparing REINFORCE with DDPO on the T2I-CompBench benchmark over three image attributes: Color, Shape, and Texture. We report average reward on unseen test set (higher is better). For each prompt, average rewards over 10 independent generated images are calculated. We report the mean and standard deviation across 3 independent runs.

| Method | Color ↑ | Shape ↑ | Texture ↑ |
|---|---|---|---|
| REINFORCE | 0.6438 (0.0132) | 0.5330 (0.0105) | 0.6359 (0.0094) |
| REINFORCE w/ BC | 0.6351 (0.0344) | 0.5347 (0.0097) | 0.6656 (0.0134) |
| DDPO | **0.6821 (0.0030)** | **0.5655 (0.0185)** | **0.6909 (0.0138)** |

We now have empirical evidence supporting the *efficiency-effectiveness trade-off* discussed in Section 3. From these results, we can conclude that fine-tuning text-to-image diffusion models is more effective with IS and clipping from PPO, or baseline corrections from REINFORCE. This bolsters our motivation for proposing LOOP as an approach to effectively combine these methods.

## 7.2 Evaluating LOOP

Next we discuss the results from our proposed RL for diffusion fine-tuning method, LOOP.

**Performance during training.** Figure 3 shows the training reward curves for different tasks, against number of epochs. LOOP outperforms DDPO (Black et al., 2023) across all seven tasks consistently throughout training. This establishes the effectiveness of sampling multiple diffusion trajectories per input prompt, and the leave-one-out baseline correction term (Eq. 9) during training. Training reward curve is smoother for the aesthetic task, as compared to tasks from the T2I-CompBench benchmark. We hypothesise that improving the attribute binding property of diffusion model is a harder task than improving the aesthetic quality of generated images.

Table 2: Comparing the performance of the proposed LOOP method with state-of-the-art baselines on the T2I-CompBench benchmark over image attributes such as Color, Shape, Texture, Spatial relation, and Numeracy. The metrics reported are the average reward on an unseen test set (higher is better). For each prompt, rewards are averaged across 10 generated images. For DDPO and LOOP, we additionally report the mean and standard deviation across 3 independent runs.

| Model | Color ↑ | Shape ↑ | Texture ↑ | Spatial ↑ | Numeracy ↑ |
|---|---|---|---|---|---|
| Stable v1.4 (Rombach et al., 2022) | 0.3765 | 0.3576 | 0.4156 | 0.1246 | 0.4461 |
| Stable v2 (Rombach et al., 2022) | 0.5065 | 0.4221 | 0.4922 | 0.1342 | 0.4579 |
| Composable v2 (Liu et al., 2022) | 0.4063 | 0.3299 | 0.3645 | 0.0800 | 0.4261 |
| Structured v2 (Feng et al., 2022) | 0.4990 | 0.4218 | 0.4900 | 0.1386 | 0.4550 |
| Attn-Exct v2 (Chefer et al., 2023) | 0.6400 | 0.4517 | 0.5963 | 0.1455 | 0.4767 |
| GORS unbiased (Huang et al., 2023) | 0.6414 | 0.4546 | 0.6025 | 0.1725 | – |
| GORS (Huang et al., 2023) | 0.6603 | 0.4785 | 0.6287 | 0.1815 | 0.4841 |
| DDPO (Black et al., 2023) | 0.6821 (0.0030) | 0.5656 (0.0185) | 0.6909 (0.0138) | 0.1961 (0.0034) | 0.5102 (0.0041) |
| LOOP ($k = 2$) | 0.6786 (0.0037) | 0.5746 (0.0125) | 0.6938 (0.0047) | 0.1801 (0.0085) | 0.5072 (0.0052) |
| LOOP ($k = 3$) | 0.7516 (0.0097) | 0.6220 (0.0173) | 0.7354 (0.0071) | 0.1966 (0.0058) | 0.5242 (0.0062) |
| LOOP ($k = 4$) | **0.7859 (0.0114)** | **0.6676 (0.0021)** | **0.7519 (0.0036)** | **0.2137 (0.0073)** | **0.5423 (0.0014)** |

Table 2 reports the average rewards on the test set across various tasks from the T2I-CompBench benchmark. For each prompt, we generate 10 different images and calculate the average rewards. LOOP consistently outperforms DDPO (Black et al., 2023) and other strong supervised learning-based baselines across all

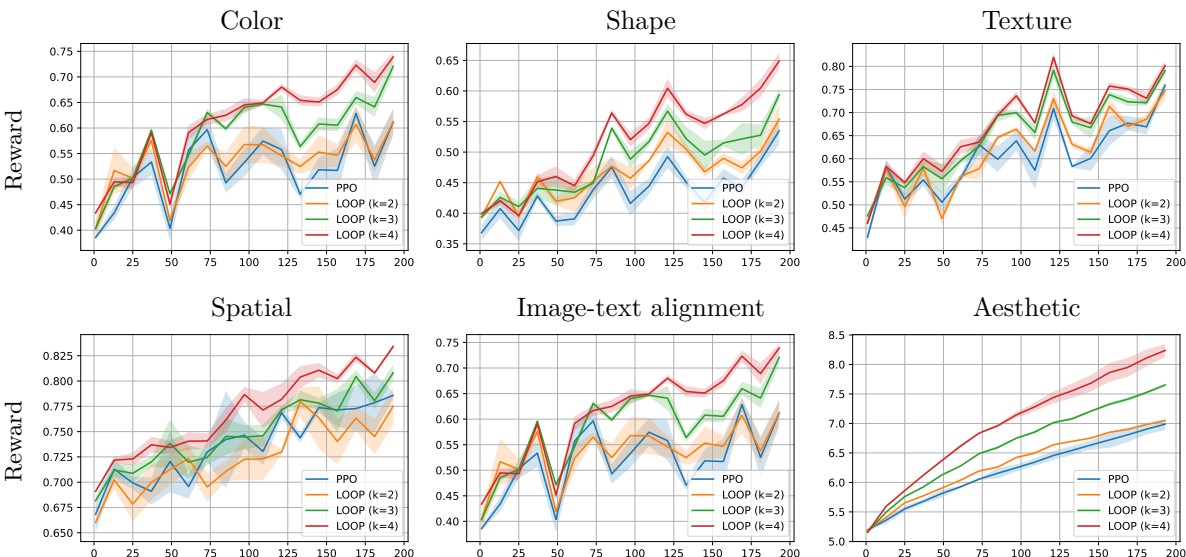

Figure 3: Comparing DDPO (referenced as PPO) with the proposed LOOP on the T2I-CompBench benchmark with respect to image attributes: **Color**, **Shape**, **Texture**, and **Spatial relationship**. We also report results on aesthetic preference and image–text alignment tasks (Black et al., 2023). The y-axis shows training reward, and the x-axis shows training epoch. Results are averaged over three independent runs; shaded regions indicate ±1 standard deviation around the mean, computed over three independent runs.

tasks. Notably, LOOP achieves relative improvements of **18.1%** and **15.2%** over DDPO on shape and color attributes, respectively.

For the aesthetic and image-text alignment objectives, the validation rewards are reported in Table 3. LOOP results in a **15.4%** relative improvement over PPO for the aesthetic task, and a **2.4%** improvement over PPO for the image-text alignment task.

Table 3: Comparing the performance of LOOP with DDPO on the aesthetic and image-text alignment tasks. Higher values are better. For each prompt, rewards are averaged across 10 generated images. We report the mean and standard deviation across 3 independent runs.

| Method | Aesthetic ↑ | Image Align. ↑ |
|---|---|---|
| DDPO (Black et al., 2023) | 6.6795 (0.0925) | 20.456 (0.4383) |
| LOOP ($k = 2$) | 6.7338 (0.0792) | 20.773 (0.4383) |
| LOOP ($k = 3$) | 7.1213 (0.0370) | 20.623 (0.0963) |
| LOOP ($k = 4$) | **7.7061 (0.1006)** | **20.912 (0.0787)** |

**Impact of number of independent trajectories ($k$).** The LOOP variant with number of independent trajectories $K = 4$ performs the best across all tasks, followed by the variant $K = 3$. This is intuitive given that Monte-Carlo estimates get better with more number of samples (Owen, 2013). Surprisingly, the performance of the variant with $K = 2$ is comparable to PPO.

# 8 Qualitative Examples

For a qualitative evaluation of the attribute-binding reasoning ability, we present some example image generations from SD, DDPO, and LOOP in Figures 1, 4, and 5.

In Figure 1 qualitative examples of the attribute binding task are presented. In the example in the first column of Figure 1, the input prompt specifies a black ball with a white cat. Stable diffusion (SD) and PPO fail to bind the color black with the generated ball, whereas LOOP successfully binds that attribute.

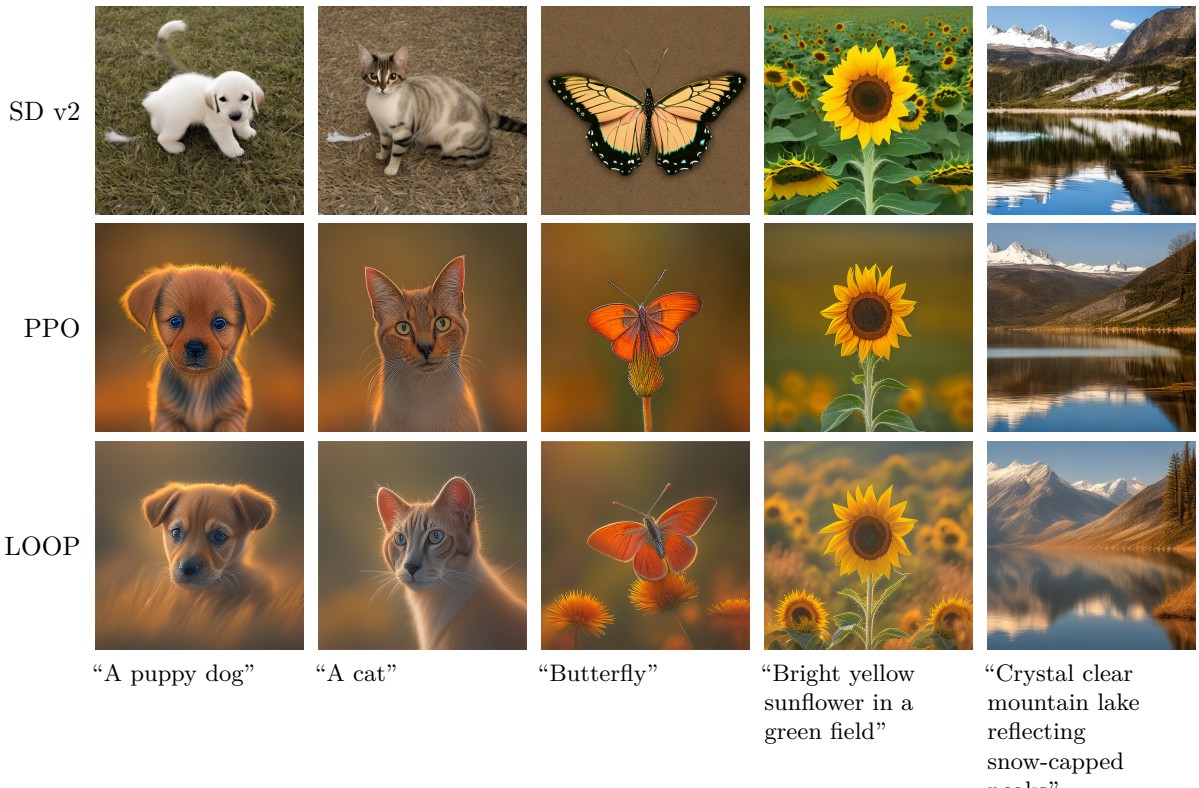

Figure 4: **LOOP improves aesthetic quality**. Qualitative examples are presented from images generated via: Stable Diffusion 2.0 (first row), PPO (second row), and LOOP $k = 4$ (third row). LOOP consistently generates more aesthetic images, as compared to PPO and SD.

Similarly, in the third column, SD and PPO fail to bind the hexagon shape attribute to the watermelon, whereas LOOP manages to do that. In the fourth column, SD and PPO fail to add the horse object itself, whereas LOOP adds the horse with the specified black color, and flowing cyan patterns.

Figure 4 highlights improvements in aesthetic quality of the generated images. Compared to SD v2 and PPO, LOOP produces sharper, more coherent compositions with balanced lighting and color tone. For example, in the second column ("a cat") and in the fourth column ("butterfly"), LOOP enhances realism and contrast while preserving overall artistic intent.

Finally, Figure 5 presents additional qualitative examples that emphasize both binding and aesthetics. LOOP accurately binds challenging color-object pairs (e.g., teal branch, pink cornfield) while producing more visually appealing and natural results. PPO and SD v2 often miss attribute alignment or produce dull, less cohesive scenes.

## 9    Conclusion

We have studied the efficiency-effectiveness trade-off between two fundamental RL methods for diffusion fine-tuning: REINFORCE and PPO. Our analysis, both theoretical and empirical, demonstrates that while REINFORCE is simpler to implement, requiring less memory and fewer hyperparameters, it suffers from high variance and poor sample efficiency compared to PPO. PPO, though more sample efficient and more effective in terms of reward optimization, comes with significant implementation overhead, requiring three models in memory simultaneously and sensitive hyperparameter tuning.

Building on these insights, we have introduced LOOP, a novel RL method for diffusion fine-tuning that combines variance reduction techniques from REINFORCE (multiple trajectory sampling and leave-one-out

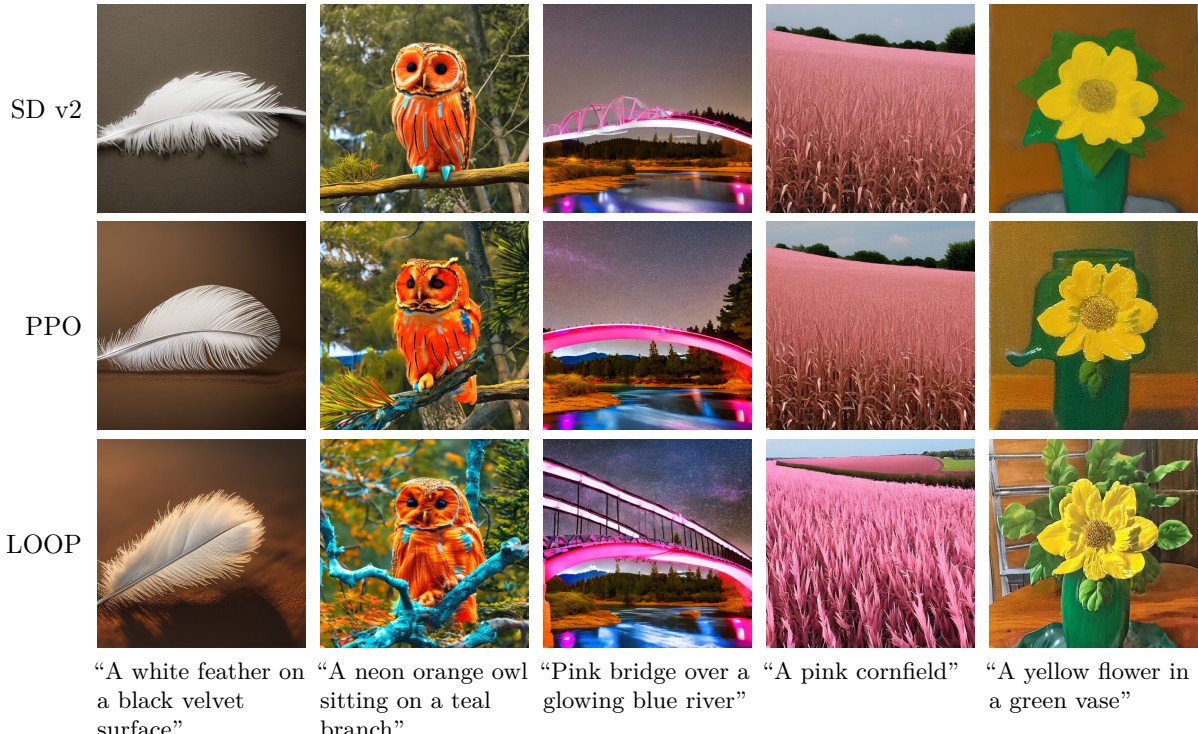

SD v2 / PPO / LOOP

"A white feather on a black velvet surface" "A neon orange owl sitting on a teal branch" "Pink bridge over a glowing blue river" "A pink cornfield" "A yellow flower in a green vase"

Figure 5: Additional qualitative examples presented from images generated via Stable Diffusion 2.0 (first row), PPO (second row), and LOOP $k = 4$ (third row). LOOP consistently generates more aesthetic images, as compared to PPO and SD (first, third, and fifth prompt). LOOP also binds the color attribute (teal branch in second example, and pink cornfield in the forth example), where SD and PPO fail.

baseline correction) with the robustness and sample efficiency of PPO (importance sampling and clipping). Our empirical evaluation on the T2I-CompBench benchmark demonstrates that LOOP achieves substantial improvements over both the base Stable Diffusion model and the state-of-the-art PPO method across multiple tasks, including attribute binding (color, shape, texture, spatial relationships), aesthetic quality, and image-text alignment.

Quantitatively, LOOP ($k = 4$) achieves substantial improvements over PPO across all evaluated tasks. On the T2I-CompBench benchmark, LOOP achieves relative improvements of **18.1**% on shape binding, **15.2**% on color binding, **8.8**% on texture binding, and **8.9**% on spatial reasoning. LOOP also improves aesthetic quality by **15.4**% and image-text alignment by **2.2**%. Qualitatively, as shown in Figures 1, 4, and 5, LOOP successfully binds attributes that previous methods fail to capture, while also producing more visually coherent and aesthetic images.

A limitation of LOOP is the increased computational cost from sampling multiple diffusion trajectories per prompt, which leads to longer training times compared to standard PPO. Future work could explore adaptive sampling strategies to reduce this overhead while maintaining LOOP's effectiveness, extend the method to other diffusion architectures and modalities, or investigate the integration of human preference modeling for better alignment with real-world objectives.

**Acknowledgements**

We thank our reviewers for their helpful feedback. This research was supported by Huawei Finland through DreamsLab, the Dutch Research Council (NWO), under project numbers 024.004.022, NWA.1389.20.183, KICH3.LTP.20.006, and VI.Veni.222.26, and the European Union under grant agreements No. 101070212 (FINDHR) and No. 101201510 (UNITE). Views and opinions expressed are those of the author(s) only and do not necessarily reflect those of their respective employers, funders and/or granting authorities.

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

# A    LOOP Algorithm Pseudo-code

This section provides a detailed pseudo-code implementation of the LOOP (Leave-One-Out PPO) algorithm for diffusion fine-tuning. The algorithm, shown in Figure 6, consists of four main steps. Step 1: sampling multiple trajectories per prompt from the old policy. Step 2: computing leave-one-out baseline advantages (Eq. 14). Step 3: updating the policy using clipped LOOP objective with importance sampling (Eq. 13). And Step 4: performing gradient descent.

**LOOP: Leave-One-Out PPO for Diffusion Fine-tuning**

```
# Input: prompts, old_policy, reward_function
# Hyperparameters: K (trajectories per prompt), epsilon (clipping)

for prompt in training_prompts:

    # Step 1: Sample K trajectories from old policy
    trajectories = []
    rewards = []
    for i in range(K):
        x_noise = sample_gaussian_noise()
        traj = diffusion_reverse_process(old_policy, prompt, x_noise)
        trajectories.append(traj)
        rewards.append(reward_function(traj.final_image, prompt))

    # Step 2: Compute leave-one-out baseline advantages
    advantages = []
    for i in range(K):
        # Baseline: average of all rewards excluding i-th reward
        baseline_i = 0
        for j in range(K):
            if j != i:
                baseline_i += rewards[j]
        baseline_i = baseline_i / (K - 1)

        # Advantage: reward minus leave-one-out baseline
        advantages.append(rewards[i] - baseline_i)

    # Step 3: LOOP update with importance sampling and clipping
    loss = 0
    for i in range(K):
        for t in range(num_timesteps):
            # Compute importance ratio
            log_prob_new = current_policy.log_prob(traj[i], t)
            log_prob_old = old_policy.log_prob(traj[i], t)
            ratio = exp(log_prob_new - log_prob_old)

            # Clipped surrogate objective
            clipped_ratio = clip(ratio, 1-epsilon, 1+epsilon)
            loss += -min(ratio * advantages[i],
                         clipped_ratio * advantages[i])

    # Step 4: Gradient descent
    optimize(loss)
```

Figure 6: Pseudo-code for LOOP. The algorithm samples $K$ trajectories per prompt, computes leave-one-out advantages $A^i = R^i - \frac{1}{K-1}\sum_{j \neq i} R^j$, and updates the policy with clipped importance sampling.

# B LOOP vs. GRPO

In this section, we compare LOOP with GRPO (Shao et al., 2024). In the introduction (Section 1), we provided a theoretical comparison of LOOP with GRPO. In this section, we provide an empirical comparison on the aesthetic task.

Figure 7 presents results from a small-scale experiment comparing GRPO with LOOP on the aesthetic fine-tuning task. We conducted this auxiliary study because the base model used in our main experiments, Stable Diffusion 2.0, has since been deprecated.[1] As a result, we ran both methods using an unofficial fork of SD 2.0.[2]

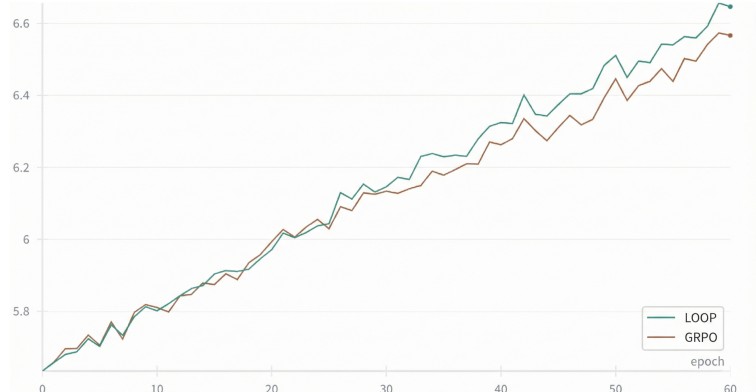

Figure 7: Comparision of LOOP vs GRPO on the aesthetic task.

The plot shows steady and stable reward improvement for both methods over training epochs. In the early phase, GRPO and LOOP exhibit closely aligned performance, indicating similar initial optimization behavior. As training progresses, LOOP consistently achieves slightly higher aesthetic rewards than GRPO, maintaining a persistent margin through convergence. Overall, while both methods demonstrate stable learning dynamics, LOOP attains better final aesthetic performance in this small-scale setting.

Overall, the figure empirically supports the claim that incorporating leave-one-out baseline correction within a PPO-style clipped objective improves optimization effectiveness for diffusion fine-tuning, yielding better alignment with aesthetic reward models while maintaining stable training dynamics.

# C Comparing Different Baseline Correction Terms

Figure 8 presents a small-scale empirical comparison between the leave-one-out baseline used in LOOP and the running-mean baseline adopted in DDPO (PPO) (Black et al., 2023), evaluated on the aesthetic task in the same toy setup used for the GRPO vs. LOOP comparison (Appendix B). Both variants demonstrate stable and monotonic reward improvement over training, confirming that either baseline strategy supports effective optimization. However, the leave-one-out baseline consistently achieves higher rewards than the running-mean baseline, with the gap becoming more pronounced in later epochs.

A possible explanation for the weaker performance of the running-mean baseline is the policy drift. Since the running mean aggregates rewards across different optimization steps, it mixes samples generated by different policies over time. As the policy evolves, older reward statistics may no longer accurately reflect the current policy distribution, leading to a misaligned baseline and suboptimal credit assignment. In contrast, the leave-one-out baseline is computed using multiple trajectories sampled from the same policy at the same iteration, making it more locally consistent and better aligned with the current optimization step. This likely contributes to the improved and more stable performance observed with the leave-one-out baseline.

---

[1]https://huggingface.co/stabilityai/stable-diffusion-2-base
[2]https://huggingface.co/Manojb/stable-diffusion-2-1-base

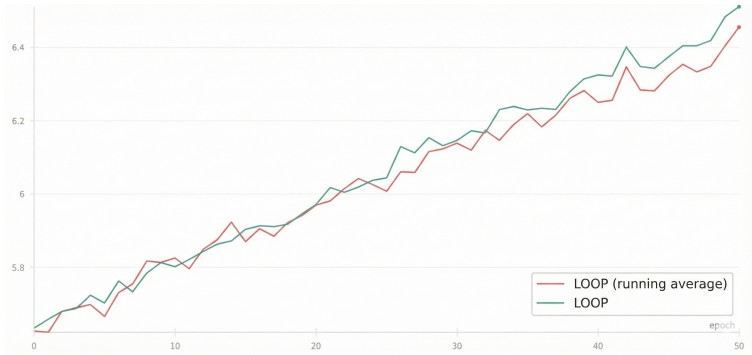

Figure 8: Comparision of LOOP vs GRPO on the aesthetic task.

# D   Importance of Clipping in PPO

Figure 9 presents an ablation study evaluating the role of clipping in PPO for diffusion fine-tuning across three T2I-CompBench tasks: Color, Shape, and Texture. We compare standard PPO with clipping against PPO without clipping under the same training configuration. Across all three tasks, PPO with clipping consistently achieves higher rewards and exhibits smoother, more stable training dynamics. While PPO without clipping initially improves, its performance is less stable and converges to lower final reward values. The gap becomes more pronounced in later epochs, indicating that clipping plays an important role in maintaining controlled policy updates and preventing excessive deviation from the reference policy. These results empirically support the theoretical motivation for clipping and demonstrate its practical importance for stable and effective diffusion model fine-tuning.

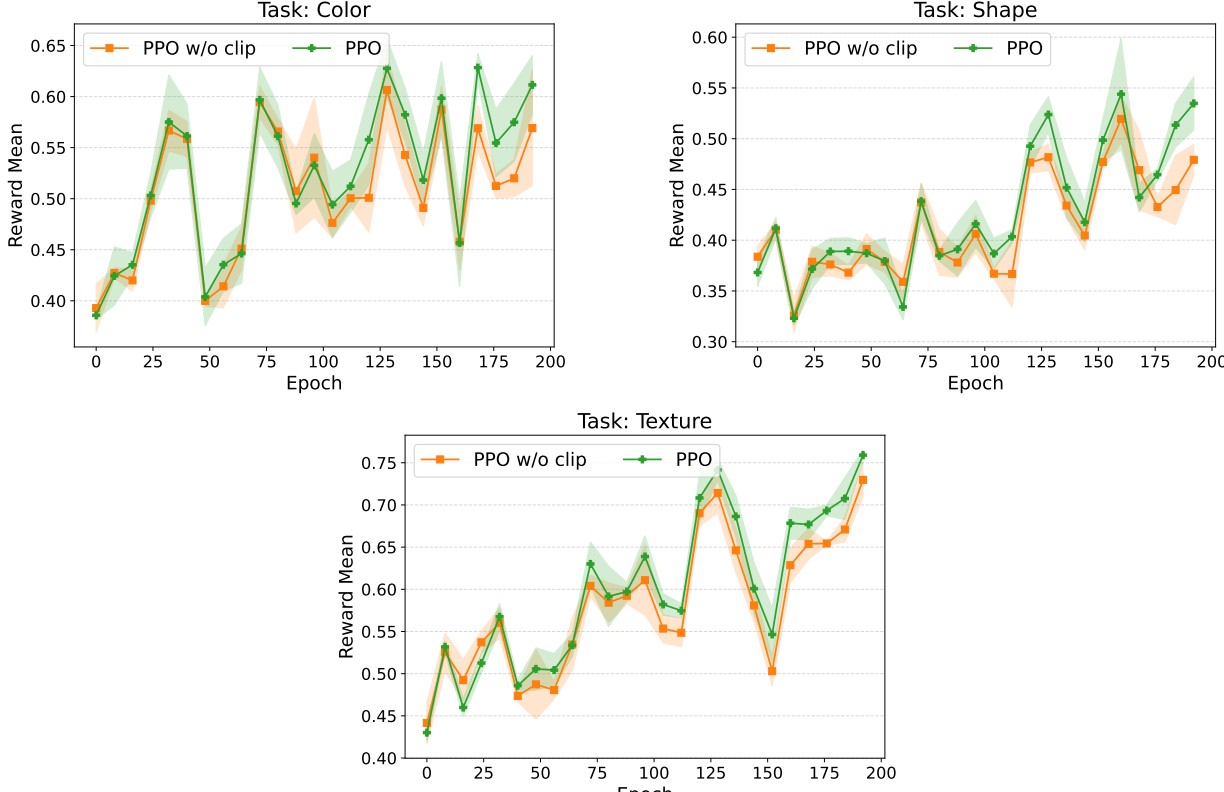

Figure 9: Effect of clipping in PPO for diffusion fine-tuning. PPO with clipping achieves more stable training and higher reward than PPO without clipping across Color, Shape, and Texture tasks.

# E  Runtime and Memory Usage

Figure 10 summarizes the computational cost of RL fine-tuning across tasks by reporting (a) mean wall-clock runtime and (b) peak GPU memory usage for PPO and LOOP with different numbers of trajectories $K$. Across tasks, runtime increases with $K$, reflecting LOOP's $O(K)$ sampling overhead from generating multiple diffusion trajectories per prompt, while PPO is consistently the fastest. Peak GPU memory remains broadly comparable across methods, with only modest increases for larger $K$, suggesting that LOOP's primary cost is additional sampling time rather than substantially higher memory footprint. Overall, the figure highlights the expected efficiency trade-off: LOOP improves sample efficiency at the expense of increased wall-clock compute, with relatively small changes in peak memory.

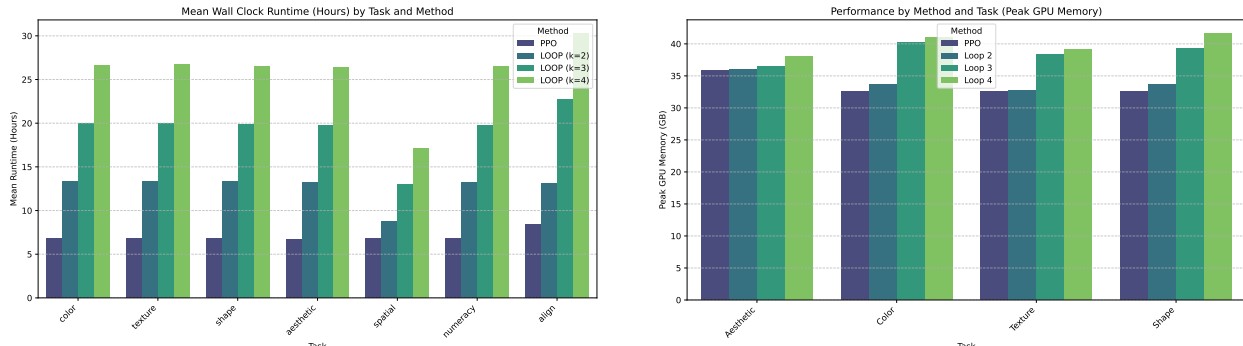

Figure 10: Runtime and memory overhead of LOOP relative to PPO across tasks. The figure on the left reports mean wall-clock runtime (hours) and the figure on the right reports peak GPU memory (GB) for each task and method variant.

# F  Reward-Diversity Trade-off

Figure 11 plots the reward (on a log-scale) against generation diversity for PPO and LOOP variants across four prompts (hippopotamus, octopus, snail, and cheetah) on the aesthetic task. Diversity is quantified as the average pairwise CLIP cosine distance within a batch of generated images, following previous work (Venkatraman et al., 2024).

Across all prompts, LOOP with $k = 4$ achieves the highest log reward while simultaneously maintaining high diversity, consistently occupying the Pareto-dominant (upper-right) region of the plot. In contrast, PPO attains the lowest diversity in every case, indicating pronounced mode collapse: optimization concentrates probability mass on a narrow subset of high-reward samples.

Increasing $k$ in LOOP improves both reward and diversity jointly, rather than trading one for the other. This suggests that sampling multiple trajectories per prompt not only strengthens optimization but also promotes exploration of distinct generation modes. The effect is consistent with the leave-one-out baseline, which reinforces relative improvements over average across diverse trajectories instead of amplifying a single dominant mode.

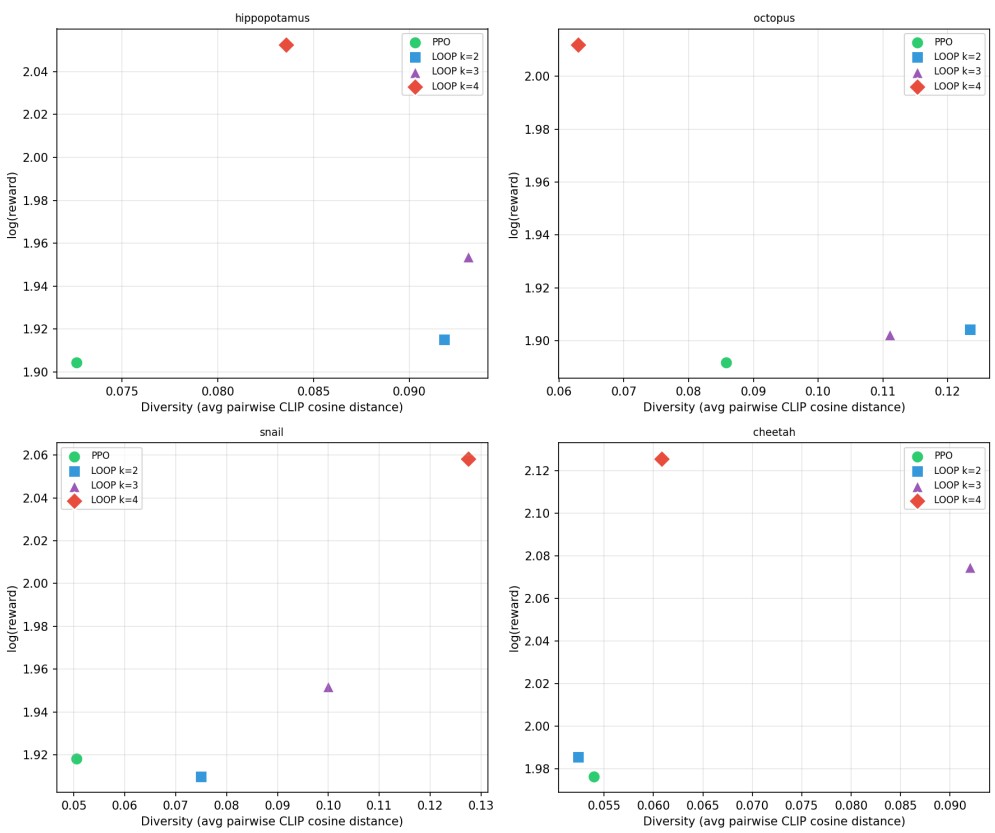

Figure 11: Reward vs. diversity trade-off for PPO and LOOP variants across four prompts on the aesthetic task. We report mean log reward and diversity, measured as the average pairwise CLIP cosine distance across a batch of generated images (Venkatraman et al., 2024). LOOP $k=4$ consistently achieves the highest log reward while maintaining competitive diversity across all prompts, whereas PPO achieves the lowest diversity scores, suggesting mode collapse towards a narrow set of high-reward images.

