# OpenReview forum: "A Simple and Effective Reinforcement Learning Method for Text-to-Image Diffusion Fine-tuning"
_TMLR — Accepted by TMLR_

### Review · Reviewer_3DSC · 2025-11-23

**Summary Of Contributions:**

This paper studies the efficiency–effectiveness trade-off between REINFORCE and PPO for RL-based post-training on text-to-image diffusion models. REINFORCE is simple and cheap, but known as suffering from high-variance and sample-inefficient; PPO is good at performance while expensive and hyperparameter-sensitive.

Based on their theoretical analysis (leveraging results from prior works like Achiam et al., 2017),  and empirical analysis, the authors propose LOOP (Leave-One-Out PPO), which combines 1) PPO’s importance sampling + clipping; 2) REINFORCE/RLOO’s multi-trajectory sampling and leave-one-out baseline variance reduction.

Experiments on T2I-CompBench (containing multiple attribute binding/image compositionality tasks), with evaluation on image aesthetics, and image-text alignment show that LOOP provides stronger performance than standard PPO (DDPO). The key claims from the paper include:
	•	LOOP consistently outperforms baselines in attribute binding (color, shape, texture, spatial).
	•	LOOP improves aesthetic and alignment scores.
	•	LOOP smooths training curves and is more stable.

**Audience:**

Yes

**Audience Explanation:**

The paper studies the efficiency–effectiveness trade-off between REINFORCE and PPO for RL-based post-training on text-to-image diffusion models. The paper leveraging results from prior works like Achiam et al., 2017, proposes a method that empirically works well  attribute binding and improves aesthetic and alignment scores.

**Broader Impact Concerns:**

The paper studies principle method for RL, thus is not  significantly concerned on the ethical implications.

**Claims And Evidence:**

Yes

**Claims Explanation:**

1. The paper provides a clean theoretical and empirical argument showing limitations of REINFORCE and PPO in diffusion RL settings, establishing a credible motivation for LOOP, using clear articulation (e.g., presentations in main text, visual aids in figures and organized results in tables)

2. The method is intuitive yet nontrivial with the following reasons:
* Multi-trajectory sampling reduces PPO variance.
* Leave-one-out baseline avoids bias introduced by simple baselines.
* Cliping + Importance Sampling to maintain stability.

3. LOOP achieves substantial improvements across all tasks:
* +18.1% (shape), +15.2% (color), +8.8% (texture) over DDPO (Table 2)
* +15.4% improvement in aesthetics (Table 3)

The improvements are significant and appear robust (K=4 best).

4. The paper’s experiments are comprehensive, including:
*  7 benchmarks (color, shape, texture, spatial, numeracy, aesthetic, alignment)
*  Multiple baselines, including PPO, stable diffusion baselines, and compositional methods.
*  3-run averages with prediction intervals.

5. The paper adapts Achiam et al.’s monotonic improvement theorem for diffusion MDPs, which strengthens its theoretical justification for PPO-style clipping in diffusion RL.

**Requested Changes:**

1. The authors reference GRPO (Shao et al., 2024) and claim conceptual similarity but do not perform empirical comparisons. Considering LOOP is extremely close to GRPO (multi-samples, leave-one-out baseline, etc.), the comparison is non-trivial in my opinion.

2. The authors only study the effect of different $k$. The proposed LOOP still involves several key components that need more careful exploration. For example
* Effect of leave-one-out
* Effect of clipping
* How does the removal of standard-deviation normalization compare with GRPO
* The sensitivity to diffusion sampler (maybe quantify the relation of stochasticity in diffusion reverse, and the number of sampling steps?)

3. As LOOP requires multi-trajectory sampling, the compute cost should be quantified. The authors  need to further study GPU hours,  memory usage etc. for comprehensive analysis, otherwise it is  hard to support the "efficiency–effectiveness" claim.

4. The experiment  settings are not detailed, which may cause misunderstanding from readers who are not fully familiar with diffusion fine-tuning. Please clarify the settings such as:
* What specific reward models are used per task?
* Are the pretrained models such as text encoder fixed or also affected by the reward?
* What sampler the diffusion model uses, do they apply ODE solver, any type of scheduler, fixed noise schedules, etc.
* How the guidance is applied in the sampling?

5. The monotonic improvement bound (Theorem 3.1) is restated from Achiam et al.

6. The evaluation does not study FiD, perceptual scores and CLIP scores.

7. Math notations are not consistent. For example, Eq. 10 uses normal font for the variables  while other places the variables use bold notations. Are they different? If so, the authors need to define them in the text.

### References

- Shao, Zhihong, Peiyi Wang, Qihao Zhu, Runxin Xu, Junxiao Song, Xiao Bi, Haowei Zhang et al. "Deepseekmath: Pushing the limits of mathematical reasoning in open language models." arXiv preprint arXiv:2402.03300 (2024).

---

> ### Author Response · Authors · 2026-01-10
> **Rebuttal to reviewer 3DSC - (1/2)**
>
> We thank the reviewer for their thorough evaluation of our work and their suggestions for improvement. We specifically address the reviewer’s feedback in what follows below, to clarify misconceptions.
>
> > The authors reference GRPO (Shao et al., 2024) and claim conceptual similarity but do not perform empirical comparisons. Considering LOOP is extremely close to GRPO (multi-samples, leave-one-out baseline, etc.), the comparison is non-trivial in my opinion.
>
> Regarding the comparison with GRPO, we quickly ran a small experiment comparing GRPO with LOOP on the aesthetic image generation task. We note that the Stable Diffusion 2 model is officially deprecated, and we found an unofficial copy of the model that has been further fine-tuned on the original official Stable Diffusion training dataset (https://huggingface.co/Manojb/stable-diffusion-2-1-base). Due to this reason, a comparison with the LOOP results in the paper (Figure 3) is not valid; we had to run LOOP again. We add the training curve here: https://anonymous.4open.science/r/TMLR-Rebuttal-4411 (GRPO v/s LOOP file). It is clear that LOOP works better than GRPO.
>
> > The authors only study the effect of different . The proposed LOOP still involves several key components that need more careful exploration. For example: 1) Effect of leave-one-out, 2) Effect of clipping, 3) How does the removal of standard-deviation normalization compare with GRPO, 4) The sensitivity to diffusion sampler (maybe quantify the relation of stochasticity in diffusion reverse, and the number of sampling steps?)
>
> Thanks for the question. Regarding the removal of the standard deviation normalization, we note that recent work has shown that removing the standard deviation term in GRPO improves its performance [1], which justifies the normalization used in LOOP. Leave-one-out serves as a baseline, and [2] shows that advantage normalization (either subtracting the mean or using z-score normalization) is critical in PPO’s performance. We believe the same would hold for LOOP. We have experiments validating the effectiveness of advantage normalization for the REINFORCE method (Fig. 2).
>
> Clipping is critical for PPO/LOOP’s performance, and we rely on results from the original PPO paper [3], which showed that clipping is essential for stable performance, and without clipping, the model training is likely to collapse. We did manage to find old experiments in which we verified this for the PPO baseline on three tasks here (https://anonymous.4open.science/r/TMLR-Rebuttal-4411 color, texture and shape files). This led to our design decision to include clipping in LOOP.
>
> For the diffusion sampler, we used the setup from the PPO for diffusion fine-tuning baseline paper [4]. We have added the diffusion sampler details in the experimental section (Section 6). We leave an ablation study with different settings of the diffusion sampler out of scope for this work and leave it to future work.
>
> [1] Liu, Zichen, et al. "Understanding r1-zero-like training: A critical perspective." arXiv preprint arXiv:2503.20783 (2025).
> [2] Xu, Shusheng, et al. "Is DPO superior to PPO for LLM alignment? A comprehensive study." arXiv preprint arXiv:2404.10719 (2024).
> [3] Schulman, John, et al. "Proximal policy optimization algorithms." arXiv preprint arXiv:1707.06347 (2017).
> [4] Black, Kevin, et al. "Training diffusion models with reinforcement learning." arXiv preprint arXiv:2305.13301 (2023).
>
>
> > The experiment settings are not detailed, which may cause misunderstanding from readers who are not fully familiar with diffusion fine-tuning. Please clarify the settings such as: What specific reward models are used per task? , Are the pretrained models such as text encoder fixed or also affected by the reward? What sampler the diffusion model uses, do they apply ODE solver, any type of scheduler, fixed noise schedules, etc. How the guidance is applied in the sampling?
>
> The details of the reward model are described in detail in the T2I benchmark’s publication [1].
>
> In Section 5, we specify that we perform a full update of the Unet backbone of the latent diffusion architecture from SD 2. Other components, such as the text encoder and VAE, are kept frozen.
>
> Regarding the diffusion model sampler, we use DDIM (Denoising Diffusion Implicit Models) as its primary sampler by default. We have added the diffusion sampler details in the experimental section (Section 6).
>
> [1] Huang, Kaiyi, et al. "T2i-compbench: A comprehensive benchmark for open-world compositional text-to-image generation." Advances in Neural Information Processing Systems 36 (2023): 78723-78747.

---

> ### Author Response · Authors · 2026-01-10
> **Rebuttal to reviewer 3DSC - (2/2)**
>
> > The monotonic improvement bound (Theorem 3.1) is restated from Achiam et al.
>
> The main theoretical contribution is in Definition 1. We show that the state transition function in diffusion models is deterministic: we use the previous action as the current state, so there is no stochastic transition. However, the state-visitation distribution is stochastic, and we define/derive the state-visitation distribution specific to the diffusion model (Eq. 11). Theorem 3.1 is re-iterated from Achiam et al. 2017. We don’t provide any proof for Theorem 3.1 and only use it to strengthen our discussion of REINFORCE v/s PPO.
>
> > The evaluation does not study FiD, perceptual scores and CLIP scores.
>
>  For FID score calculation, we need access to the ground-truth image distribution. In this setup, we only have access to training/eval prompts and an evaluation pipeline, without ground-truth images, so it’s not feasible to compute FID scores. The perceptual scores reflect, in a way, the aesthetic properties of the image, as captured in the aesthetic task in our experiments. CLIP scores reflect the image-text alignment score, as measured in the image-text alignment task.
>
> > Math notations are not consistent. For example, Eq. 10 uses normal font for the variables while other places the variables use bold notations. Are they different? If so, the authors need to define them in the text.
>
> Thanks for noticing. We have fixed the notations in the draft.

---

> > ### Comment · Reviewer_3DSC · 2026-01-12
> > **Response to the authors**
> >
> > Dear authors,
> >
> > Thank you for providing responses to my questions. I think most of them have addressed my concerns. There is one remaining point that I may not fully understand, and I would appreciate further clarification.
> >
> > Regarding the FID evaluation, I understand that you only have access to (and only use) the prompts during training, not the images. However, one non-trivial setting I have in mind is a zero-shot evaluation in which you use the evaluation prompts to generate images and then compute FID against the corresponding evaluation image set. For example, a common protocol on MS-COCO uses the 30k validation prompts to generate 30k images and then computes FID using the validation images as the reference distribution. You may refer to GigaGAN’s evaluation protocol for details.
> >
> > Do you think this setting is feasible for your evaluation? I believe it could strengthen the contribution by aligning your evaluation with common text-to-image settings, and by demonstrating that the fine-tuned model remains within the same image manifold.

---

> ### Author Response · Authors · 2026-01-12
> **Response to the reviewer**
>
> Dear Reviewer,
>
> Thanks for your prompt response. We are glad that our replies have addressed most of your concerns.
>
> Regarding FID score evaluation, we believe it makes more sense to evaluate FID in an offline reinforcement learning fine-tuning setting (DPO-style methods), where we have access to datasets of the form (prompt, preferred image, rejected image). In that setting, one can meaningfully compare the distribution of generated images with the distribution of ground-truth preferred images. For example, Diffusion-RPO [1] reports FID scores (Table 2), since a reference preference image distribution is available in an offline fine-tuning setting.
>
> In contrast, in online RL-based diffusion fine-tuning, it is not standard practice to report FID scores due to the absence of a well-defined ground-truth image distribution [2,3,4].
>
> Moreover, FID primarily measures distributional similarity to natural images, and improvements in task-specific objectives, such as aesthetic preference optimized via a reward model trained on human judgments, do not necessarily correlate with FID. In particular, stylized or artistically enhanced images can yield worse FID scores despite being preferred by humans.
>
> This phenomenon is empirically demonstrated by Jayasumana et al. [5], who show that models with lower (better) FID are not necessarily preferred by human evaluators, while models with higher FID can be favored in terms of perceived image quality (Table 3), highlighting a mismatch between FID and human-centric quality assessment.
>
>
> Ref:
>
> [1] Gu, Yi, et al. "Diffusion-rpo: Aligning diffusion models through relative preference optimization." arXiv preprint arXiv:2406.06382 (2024).
>
> [2] Black, Kevin, et al. "Training diffusion models with reinforcement learning." arXiv preprint arXiv:2305.13301 (2023).
>
> [3] Fan, Ying, et al. "Dpok: Reinforcement learning for fine-tuning text-to-image diffusion models." Advances in Neural Information Processing Systems 36 (2023): 79858-79885.
>
> [4] Clark, Kevin, et al. "Directly fine-tuning diffusion models on differentiable rewards." arXiv preprint arXiv:2309.17400 (2023).
>
> [5] Jayasumana, Sadeep, et al. "Rethinking fid: Towards a better evaluation metric for image generation." Proceedings of the IEEE/CVF Conference on Computer Vision and Pattern Recognition. 2024.

---

> > ### Comment · Reviewer_3DSC · 2026-01-17
> > **Thanks for the clarification**
> >
> > Dear authors,
> >
> > Thank you for the clarification. I have read the references and agree with your claims above. My motivation of further evaluation was to see how much we shrink the diversity as RL finetuning intrinsically enhances the mode-seeking of the model. However, like you said, FID may not be a good metric to fully reflect this property. I appreciate your response and will finalize my recommendation.

---

> > > ### Comment · Reviewer_3DSC · 2026-01-17
> > > **Regarding computation**
> > >
> > > Dear authors,
> > >
> > > Can you improve your revision regarding the computation cost part? I see your revision in acknowledging the computation cost increases, but we need to be transparent to readers by quantifying how much computation is increased. Thanks!

---

> > > > ### Author Response · Authors · 2026-01-21
> > > > **Response regarding computation cost discussion**
> > > >
> > > > Dear Reviewer,
> > > >
> > > > Thank you for your feedback.
> > > >
> > > > We have addressed your concern by revising the discussion of LOOP's computational cost in the introduction (highlighted in blue). The revised text explicitly acknowledges that LOOP incurs O(K) additional forward passes per prompt compared to PPO and outlines potential mitigation strategies inspired by distributed RL training techniques for foundation models [1,2].
> > > >
> > > > Ref:
> > > >
> > > > [1] Han, Zhenyu, et al. "AsyncFlow: An Asynchronous Streaming RL Framework for Efficient LLM Post-Training." arXiv preprint arXiv:2507.01663 (2025).
> > > >
> > > > [2] Bartoldson, Brian, et al. "Trajectory balance with asynchrony: Decoupling exploration and learning for fast, scalable llm post-training." arXiv preprint arXiv:2503.18929 (2025).

---

> > > > > ### Comment · Reviewer_3DSC · 2026-01-21
> > > > >
> > > > > Thank you. I understand it incurs O(K) additional forward passes and have read your revision. However, I still think the empirical justification (and more accurate evidence than magnitude estimation) is needed to help understand the cost and how LOOP improves sample efficiency. That's why I proposed **further study GPU hours, memory usage etc. for comprehensive analysis** in my original review and this is also suggested by other reviewers. Could you please let us know if there is any reason that you think it is not necessary?

---

### Review · Reviewer_yJgL · 2025-12-02

**Summary Of Contributions:**

The paper studies reinforcement learning (RL)–based fine-tuning of text-to-image diffusion models under black-box rewards, with a particular focus on the trade-off between simple REINFORCE-style policy gradients and Proximal Policy Optimization (PPO). The authors first formalize diffusion sampling as an MDP and re-derive the PPO objective in this setting, along with a diffusion-specific statement of the monotonic policy-improvement bound. They then empirically compare vanilla REINFORCE, REINFORCE with a baseline, and PPO (DDPO) across several reward functions (attribute binding tasks from T2I-CompBench, aesthetics, and image–text alignment), showing that PPO plus baselines improves training stability and final reward over REINFORCE variants. Building on this analysis, they introduce Leave-One-Out PPO (LOOP), which augments PPO with (i) sampling K>1 trajectories per prompt and (ii) a leave-one-out baseline $b_i$ when computing advantages, while retaining importance sampling and clipping. LOOP is shown to outperform DDPO and several non-RL baselines on T2I-CompBench (color, shape, texture, spatial, numeracy) and on aesthetic and image–text alignment rewards, with the strongest results for K=4. Qualitative examples suggest improved attribute binding and aesthetics compared to Stable Diffusion v2 and DDPO.


Key strengths:
1. Clear problem formulation: the paper gives a concise background on diffusion-as-MDP and shows how PPO objectives map to this setting.
2. Practical, incremental algorithm: LOOP is conceptually simple (RLOO + PPO) and easy to implement on top of existing DDPO codebases.
3. Strong empirical gains: consistent improvements over DDPO and previous compositionality methods on T2I-CompBench and on aesthetic / alignment tasks, including sizable relative improvements (≈15−18%) on shape/color binding.
4. Ablation over K: the study of K=2,3,4 provides some insight into the effect of multiple trajectories per prompt and connects to Monte-Carlo variance reduction.

Key weaknesses:
1. Despite the stated focus on an “efficiency–effectiveness trade-off”, the paper does not provide quantitative compute / memory / sample-efficiency comparisons (e.g., wall-clock time, GPU memory, number of reward evaluations) between REINFORCE, PPO, and LOOP.
2. The theoretical part largely reuses known results and adapts them to the diffusion setting; the novelty of the theory is modest and not clearly separated from prior work.
3. LOOP still requires a reference policy and a reward model, and also multiple trajectories per prompt; it is not obvious from the experiments that it is more efficient than PPO for a fixed compute budget.
4. Limited baselines: there is no comparison to newer RL-style diffusion methods (e.g., DPO-style, Diffusion-RPO, or GRPO-inspired variants), and even for GRPO, the authors explicitly avoid a head-to-head comparison while acknowledging conceptual similarity.
5. Evaluation is restricted to Stable Diffusion v2 and specific reward models; generalization to other diffusion architectures / resolutions / domains is not demonstrated.
6. Some claims (e.g., “first systematic study of REINFORCE vs PPO for diffusion,” or “KL regularization has minimal effect”) rest mainly on a narrow set of experiments and prior citations rather than broad empirical evidence within this work. In particular, removing the KL term can lead to significant reduction in the diversity of generation which is undesirable. (see [1][2]) It is necessary to show how the current method effects the sample diversity empirically.

[1] S Venkatraman et. al. Amortizing intractable inference in diffusion models for vision, language, and control
[2] Y Fan et. al. DPOK: Reinforcement Learning for Fine-tuning Text-to-Image Diffusion Models

**Additional Comments:**

Overall, I find the paper to be practically useful contribution with very little theoretical novelty. The empirical section is the main strength; which however looks incomplete right now. Tightening the claims around efficiency and adding a few targeted ablations would substantially improve the paper.
From a reproducibility standpoint, releasing code and training scripts for LOOP (or at least pseudo-code plus detailed hyperparameters) would make it much easier for practitioners to adopt the method.

**Audience:**

Yes

**Audience Explanation:**

TMLR’s readership includes researchers working on diffusion models, RLHF/RL-from-feedback, and generative modeling more broadly. This paper sits at the intersection of these topics and addresses a very practical question: how should we fine-tune diffusion models under black-box rewards in a way that is both performant and not overly fragile?
More specifically:
1. RL-based fine-tuning of generative models (LLMs or diffusion) is currently a very active area; a method like LOOP, which is simple and empirically strong, is likely to be of practical interest to people building aligned T2I systems.
2. The adaptation of RLOO-style variance reduction to diffusion PPO is conceptually straightforward yet non-trivial to implement correctly; a well-documented version with empirical validation is useful to practitioners.
3. The focus on attribute binding (T2I-CompBench) addresses a widely-recognized weakness of current text-to-image models, so improvements here are directly relevant to many applications.
4. Even if the theoretical contribution is modest, the empirical insight that “PPO-like methods benefit significantly from multiple trajectories + leave-one-out baselines in diffusion” is a concrete design lesson.
In short, this is an incremental but practically meaningful contribution that would interest the segment of TMLR’s audience working on diffusion models, RL for generative models, and text-to-image alignment.

**Broader Impact Concerns:**

The paper focuses on RL-based fine-tuning of text-to-image diffusion models, improving attribute binding and aesthetic quality. This has the usual dual-use implications of stronger generative models:

1. Positive: Better alignment with prompts can improve accessibility (e.g., clearer visualizations, educational material), creative tools, and downstream applications like design or scientific illustration.

2. Negative: Improved controllability and aesthetics can also make it easier to generate persuasive misinformation, deepfakes, or harmful content that more faithfully matches malicious prompts.

I do not see novel ethical risks unique to LOOP beyond those already present with text-to-image diffusion and RL-based fine-tuning, but the paper would benefit from a short, explicit discussion of these issues.

**Claims And Evidence:**

Yes

**Claims Explanation:**

For the core empirical claim—that LOOP achieves better reward than DDPO/PPO and REINFORCE-style baselines on T2I-CompBench, aesthetics, and image–text alignment—the evidence is generally convincing:
The experimental setup is described with reasonable detail (base model, reward functions, optimization hyperparameters).
Training curves (Figures 2–3) and test-set tables (Tables 1–3) consistently show LOOP outperforming DDPO and REINFORCE variants on the reported metrics.
The effect of multiple trajectories K is documented and aligns with the Monte-Carlo variance-reduction story (performance improves with larger K, especially moving from K=2 to 4).

However, several secondary or stronger phrased claims are only partially supported:
1. The paper emphasizes an efficiency–effectiveness trade-off and suggests that LOOP “achieves a better balance between computational efficiency and performance,” but there are no measurements of wall-clock time, GPU hours, peak memory, or number of reward calls per achieved reward level. Without such metrics, one can only conclude that LOOP is more effective, not that it is more efficient than PPO for a fixed budget.
2. The theoretical section adapts known monotonic improvement bounds to the diffusion setting but does not empirically validate the specific bound (e.g., by measuring TV distances or verifying the bound’s tightness); the main role of this section is explanatory, not predictive. I think it might be better for the author's to clarify their contributions there or remove certain things from that section and focus on what additional novelty this work provides.
3. The work leans on prior empirical observations to justify omitting KL regularization and standard-deviation normalization, but within this paper those design choices are not ablated, so the reader cannot tell how crucial they are in the current setting.

Overall, I would say the central algorithmic and empirical claims are well-supported, while claims about efficiency, generality, and design choices are somewhat under-evidenced and would benefit from additional ablations and metrics. As highlighted before, removing the KL term can lead to significant reduction in the diversity of generation which is undesirable. (see [1][2]) Author should show how the current method effects the sample diversity empirically. Based my personal experience of training models with and without KL term for diffusion fine-tuning, the high rewards/improvements can come at the cost of mode collapse of the sampling distribution.

**Requested Changes:**

important for acceptance;
1. Quantify the efficiency–effectiveness trade-off.
 Provide comparisons of compute and sample efficiency between REINFORCE, REINFORCE+baseline, PPO, and LOOP for at least one task.
Suggested metrics: wall-clock time to reach a target reward, total number of reward model evaluations, GPU hours, and peak GPU memory. This is essential to support the framing that LOOP offers a better balance between efficiency and performance.
2. Clarify what is theoretically novel vs prior work.
The monotonic policy improvement bound closely follows Achiam et al.; please clearly state what is formally new and what is a direct specialization of existing theorems.
Consider moving some of the proof details to an appendix while focusing the main text on the intuition and the implications specific to diffusion.
3. Ablations on LOOP design choices.
Add ablations isolating the contributions of (a) multiple trajectories K, (b) leave-one-out baseline vs simple mean baseline vs no baseline, and (c) clipping vs no clipping, ideally on at least one T2I-CompBench task and one aesthetic/alignment task.
This would help demonstrate that the combination proposed in Eq. (13–14) is actually necessary, rather than any one component being sufficient.
4. Clarify and, if possible, justify the “efficiency” claim for LOOP.
As written, LOOP still uses a reference policy, a current policy, and a reward model—like PPO—while also sampling multiple trajectories per prompt. On its face this seems more expensive than DDPO.
Either (a) show that LOOP reaches higher reward in fewer optimization steps or with less compute, or (b) revise the narrative to emphasize performance and stability rather than computational efficiency.
5. Strengthen or reframe the relationship to GRPO and other contemporary methods.
Since the method is “conceptually similar” to GRPO, the absence of any empirical comparison is a noticeable gap. If possible, include at least a small-scale comparison or justification why a direct comparison is infeasible (e.g., implementation issues, domain mismatches).
Also consider situating LOOP relative to more recent diffusion-alignment methods (e.g., Diffusion-RPO, DPO-style approaches), even if only at the level of discussion and limitations.
6. Plot the diversity of samples. As the KL term has been removed it'll make sense to report the sample diversity of the RL-finetuned model for sanity check whether the improvements come at the cost of mode collapse around high reward samples.
7. Expand experimental coverage or explicitly narrow the claims.
All experiments use Stable Diffusion v2; the claims sometimes read more generally (“diffusion models” in general). Either (a) add results on at least one additional base model (e.g., SD v1.5 or a different latent diffusion architecture) or (b) soften the claims to “for SD v2 under the studied rewards” and discuss potential generalization.

Would strengthen the work but not strictly required:
1. Clarify reward models and evaluation protocol. [Optional]
Provide more detail on the reward networks used for T2I-CompBench, aesthetic, and image–text alignment tasks (architectures, training data, normalization).
Clarify how training/validation/test splits are constructed for each task and whether any prompts overlap with the pre-training data of SD v2 or the reward models.
2. Hyper-parameter sensitivity and robustness. [Optional]
PPO is known to be hyper-parameter sensitive; it would be useful to see whether LOOP is comparably sensitive or more robust (e.g., to learning rate, clipping parameter ϵ, and K).
Even a small study varying such hyperparameters would add credibility to the “simple and effective” claim.

---

> ### Author Response · Authors · 2026-01-10
> **Rebuttal to Reviewer yJgL (1/2)**
>
> We thank the reviewer for their thorough evaluation of our work and their suggestions for improvement. Below, we specifically address the reviewer’s feedback in what follows below, to clarify misconceptions.
>
> > Quantify the efficiency–effectiveness trade-off. Provide comparisons of compute and sample efficiency between REINFORCE, REINFORCE+baseline, PPO, and LOOP for at least one task. Suggested metrics: wall-clock time to reach a target reward, total number of reward model evaluations, GPU hours, and peak GPU memory. This is essential to support the framing that LOOP offers a better balance between efficiency and performance.
>
> We would like to clarify that the efficiency component in this work refers to sample efficiency. For the same number of input prompts, LOOP achieves much better performance than PPO or other baselines. We are not claiming improvement in training efficiency here. Since we are sampling multiple trajectories, we expect the training time to be slower than PPO. We leave it as future work to study and improve the training efficiency of LOOP. We have clarified the distinction between sample efficiency and computational efficiency in the revised draft's introduction (in blue text).
>
> > Clarify what is theoretically novel vs prior work. The monotonic policy improvement bound closely follows Achiam et al.; please clearly state what is formally new and what is a direct specialization of existing theorems. Consider moving some of the proof details to an appendix while focusing the main text on the intuition and the implications specific to diffusion
>
> The main theoretical contribution is in Definition 1. We show that the state transition function in diffusion models is deterministic: we use the previous action as the current state, so there is no stochastic transition. However, the state-visitation distribution is stochastic, and we define/derive the state-visitation distribution specific to the diffusion model (Eq. 11). Theorem 3.1 is re-iterated from Achiam et al. 2017. We don’t provide any proof for Theorem 3.1 and only use it to strengthen our discussion of REINFORCE v/s PPO.
>
> > Ablations on LOOP design choices. Add ablations isolating the contributions of (a) multiple trajectories K, (b) leave-one-out baseline vs simple mean baseline vs no baseline, and (c) clipping vs no clipping, ideally on at least one T2I-CompBench task and one aesthetic/alignment task. This would help demonstrate that the combination proposed in Eq. (13–14) is actually necessary, rather than any one component being sufficient.
>
> We present results for different K values in Figure 3 for training and in Table 2 for the test set. The reason for choosing a leave-one-out baseline is that using a simple mean as the baseline yields a biased estimator, whereas a leave-one-out baseline yields an unbiased estimator [1,2]. Since a biased estimator yields an incorrect estimate of the true estimator in expectation, we did not experiment with it.
>
> [1] https://swift.readthedocs.io/en/latest/Instruction/GRPO/AdvancedResearch/RLOO.html
> [2] Kool, Wouter, Herke van Hoof, and Max Welling. "Buy 4 reinforce samples, get a baseline for free!." (2019)
>
>
> > Clarify and, if possible, justify the “efficiency” claim for LOOP. As written, LOOP still uses a reference policy, a current policy, and a reward model—like PPO—while also sampling multiple trajectories per prompt. On its face, this seems more expensive than DDPO. Either (a) show that LOOP reaches a higher reward in fewer optimization steps or with less compute, or (b) revise the narrative to emphasize performance and stability rather than computational efficiency.
>
> Thanks for the question. As we noted earlier, we refer to sample efficiency here, not training efficiency. Figure 3 shows that LOOP (K=3,4) achieves higher performance than PPO from the very start of the optimization process. We have clarified the distinction between sample efficiency and computational efficiency in the revised draft's introduction (in blue text).

---

> ### Author Response · Authors · 2026-01-10
> **Rebuttal to reviewer yJgL (2/2)**
>
> >  Strengthen or reframe the relationship to GRPO and other contemporary methods. Since the method is “conceptually similar” to GRPO, the absence of any empirical comparison is a noticeable gap. If possible, include at least a small-scale comparison or justification for why a direct comparison is infeasible (e.g., implementation issues, domain mismatches). Also consider situating LOOP relative to more recent diffusion-alignment methods (e.g., Diffusion-RPO, DPO-style approaches), even if only at the level of discussion and limitations.
>
> Regarding the comparison with GRPO, we quickly ran a small experiment comparing GRPO with LOOP on the aesthetic image generation task. We note that the Stable Diffusion 2 model is officially deprecated, and we found an unofficial copy of the model that has been further fine-tuned on the original official Stable Diffusion training dataset (https://huggingface.co/Manojb/stable-diffusion-2-1-base). Due to this reason, a comparison with the original LOOP results in the paper (Figure 3) is not valid. We add the training curve here: https://anonymous.4open.science/r/TMLR-Rebuttal-4411 (see the GRPO_vs_LOOP.png file). It is clear that LOOP works better than GRPO
>
> Regarding a comparison with the DPO-style approach, since DPO is an offline learning method, whereas LOOP/GRPO/PPO are all online, we did not include DPO-style approaches in the paper. We have added a discussion around these methods in Section 2.5 (in blue text).
>
>
> > Plot the diversity of samples. As the KL term has been removed, it'll make sense to report the sample diversity of the RL-finetuned model for a sanity check whether the improvements come at the cost of mode collapse around high reward samples.
>
> We are not sure what the question is here. We present qualitative examples in Figures 1 and 5 from LOOP and GRPO. Please note that we did not choose these samples based on their reward. We made sure to add interesting, different and random prompts, generated images from the models, and included them here.
>
>
> > Expand experimental coverage or explicitly narrow the claims. All experiments use Stable Diffusion v2; the claims sometimes read more generally (“diffusion models” in general). Either (a) add results on at least one additional base model (e.g., SD v1.5 or a different latent diffusion architecture) or (b) soften the claims to “for SD v2 under the studied rewards” and discuss potential generalization.
>
> Thanks for the question. We note that most diffusion fine-tuning papers pick a single model and show improvements on that [1,2,3]. We will make sure to highlight in the draft that the results are valid for SD v2 under the studied rewards in the draft.
>
> [1] Fan, Ying, et al. "Dpok: Reinforcement learning for fine-tuning text-to-image diffusion models." Advances in Neural Information Processing Systems 36 (2023): 79858-79885.
> [2] Black, Kevin, et al. "Training diffusion models with reinforcement learning." arXiv preprint arXiv:2305.13301 (2023).
> [3] Wallace, Bram, et al. "Diffusion model alignment using direct preference optimization." Proceedings of the IEEE/CVF Conference on Computer Vision and Pattern Recognition. 2024.

---

> ### Author Response · Authors · 2026-01-21
> **Psuedo-code for LOOP**
>
> Dear reviewer,
>
> As per your comment, we have added a pseudo-code for LOOP method's implementation in Appendix A of the draft.

---

> > ### Comment · Reviewer_yJgL · 2026-02-01
> > **Response to authors**
> >
> > Dear authors,
> > Thanks for responding to my questions and apologies for the delay in my response.
> >
> > However, I still have concerns regarding computational efficiency and would like to see some wall clock time and FLOPs per training step comparison with DPPO or DPOK. For diffusion finetuning, sample efficiency is not really a bottleneck (unlike certain real-world simulators) as we just need more prompts and sampling steps to generate more trajectories. It might very well be possible that even though PPO requires more sampling to get to the same reward, it might be overall cheaper in terms of compute to get there. Hence, I am not completely convinced that LOOP is a more effective algorithm for Diffusion Finetuning.
> > Moreover, the authors have not addressed the concerns I have raised in weakness number 6, where I highlighted that removing the KL term can lead to a significant reduction in the diversity of generation, which is undesirable.  It is necessary to show how the current method affects the sample diversity empirically. In particular, can the authors provide experiments similar to Figure 3 of this work [1] where multiple samples are generated per prompt, and avg reward and sample diversity are reported? This is important as the higher reward in LOOP might come at the cost of sample diversity, and the model is overfitting to certain modes of the distribution. (especially since there is no KL term)
> >
> > [1] S Venkatraman et. al. Amortizing intractable inference in diffusion models for vision, language, and control

---

### Review · Reviewer_J2GG · 2025-12-13

**Summary Of Contributions:**

Please excuse the very belated review.

The paper presents a study on RL based finetuning of text-to-image diffusion models. The authors evaluate different design decisions common to policy gradient methods, such as multiple trajectory sampling and clipping. They present results that show that improved performance can be obtained from the specific combination of multiple rollouts and leave-one-out baseline design common to modern REINFORCE variations such as RLOO (an LLM method) and PPO style clipping. They term this combination of design decisions LOOP.

**Audience:**

Yes

**Audience Explanation:**

While the methods are well established in other parts of the literature, I believe there is merit and potential interest in seeing these insights tested and replicated in other parts of the literature.

**Claims And Evidence:**

No

**Claims Explanation:**

Note that overall the claims made in the paper are well supported. However, I believe one core part of the claim in the paper is missing, which is going to be the major focus of the method.

While PPO does introduce the clipping mechanism to prevent large importance ratios, there is also the design dimension of the value baseline. While the LLM community is currently debating the merits of trained value functions vs multi-trajectory averages, there are several possible ways to introduce the baseline/value estimator into the algorithm. The LOOP method not only uses clipping plus multiple rollouts, it also uses a leave one out baseline. If I understand correctly, the main prior method uses (DDPO) uses a running mean and std deviation. So the presented algorithm changes multiple design choices, but does not fully evaluate where performance improvements come from. As leave-one-out is relevant enough to be chosen as the title of the method, I think this merits investigation.

As all of the presented design decisions are common practice in the literature, and none require novel exposition or additional theory, I think it is reasonable to ask for the comparison between design choices to be expanded. While all the results are technically correct, the paper boils down to a hyperparameter tuning over the number of rollouts for policy gradients in diffusion. TMLR specifically only asks that claims are substantiated, but I think in this case it is reasonable to also request thoroughness given the relative simplicity of the contribution, and the likely interest of the community in the question which parts of the changed algorithm are relevant for the performance increase.

Some additional minor questions/nitpicks:

How are the error bars estimated to be at 80% prediction interval if only three runs were trained? What method and statistical assumptions lead to an 80% prediction intervals?

The results in Figure 2 and Figure 3 do not seem to align. The PPO method evaluated in Figure 2 seems to reach substantially different returns as the PPO method in Figure 3. i might have missed the differences, but I think the naming should be distinct if these are distinct methods. The table is again inconsistent with the numbers in the Figure. If the table is on a held-out set, it is also strange why the results would be better by a decent margin than those achieved on the training set (compare Figure 2 Color PPO with Table 1 Color DDPO).

**Requested Changes:**

Please add a comparison between the running mean and std deviation used as the baseline in DDPO, and the leave-one-out baseline. if possible, comparing against a trained value function would be even better, but since this seems to be non-standard in the related work, I accept if the authors declare this as out-of-scope.

Please clarify the statistical methodology and explain the different results in different figures.

Please add meaningful confidence intervals to the tables as well for clarity.

---

> ### Author Response · Authors · 2026-01-10
> **Rebuttal to Reviewer J2GG**
>
> We thank the reviewer for their thorough evaluation of our work and their suggestions for improvement. We specifically address the reviewer’s feedback in what follows below, to clarify misconceptions.
>
> > Please add a comparison between the running mean and std deviation used as the baseline in DDPO, and the leave-one-out baseline. if possible, comparing against a trained value function would be even better, but since this seems to be non-standard in the related work, I accept if the authors declare this as out-of-scope.
>
> In LOOP, we sample multiple (K) trajectories (reverse diffusion process) for each given prompt, and the main idea is that the baseline value should be based on samples from the K trajectories, in a similar spirit to the REINFORCE policy gradient [1]. Therefore, using a running mean and std. deviation does not make sense here.
>
> [1] Kool, Wouter, Herke van Hoof, and Max Welling. "Buy 4 reinforce samples, get a baseline for free!." (2019).
>
> > Please clarify the statistical methodology and explain the different results in different figures. How are the error bars estimated to be at 80% prediction interval if only three runs were trained? What method and statistical assumptions lead to an 80% prediction intervals?
>
> Thanks for pointing out the difference between Figures 2 and 3 for LOOP. There was a small error in the plotting function for the figures, which we have now fixed and updated in the draft. Regarding the confidence interval, we have now plotted mean +/- std. deviation. As you pointed out, since we have access to only 3 runs, calculating a 80% confidence interval does not make sense, so we have updated it. The results in the table are evaluated on a separate hold-out test set; the values are different.
>
> > Please add meaningful confidence intervals to the tables as well for clarity.
>
> We have also added std. deviation in the test-set results shown in the tables.

---

> > ### Comment · Reviewer_J2GG · 2026-01-11
> > **Value baseline**
> >
> > > using a running mean and std. deviation does not make sense here
> >
> > I am unsure why it would not make sense? A k-sample based baseline could potentially have a very high variance compared to other methods.

---

> > > ### Author Response · Authors · 2026-01-13
> > > **Response to reviewer J2GG**
> > >
> > > Thanks for your response.
> > >
> > > We ran an experiment where we replaced leave-one-out baseline with a running mean and std. deviation baseline term, similar to DDPO.
> > >
> > > The results are added here: https://anonymous.4open.science/r/TMLR-Rebuttal-4411/LOOP_RunAvg_vs_LOOP.png. The leave-one-out baseline results in a better performance. A likely explanation is baseline mismatch: the running-average baseline aggregates value estimates obtained under earlier policies and potentially different reward distributions, which can become stale and misaligned with the current policy. This mismatch leads to a noisier and biased advantage signal. In contrast, the leave-one-out baseline used in LOOP is computed from the current on-policy samples, ensuring better estimate of the value function under the current policy and yielding a more reliable advantage estimate.

---

### Author Response · Authors · 2026-01-21
**Summary of Reviewer Discussion and Draft Revisions**

We sincerely thank all reviewers for their thoughtful reviews and constructive feedback. We have uploaded a revised version of the paper with all textual changes highlighted in blue. Below, we summarize the main revisions addressed in the updated manuscript and during our discussion with the reviewers:

- Clarification of efficiency terminology (highlighted text in the introduction): We have clarified that the efficiency component discussed in this work refers specifically to sample efficiency, and we explicitly state that we are not claiming improvements in training computational efficiency.
- Additional references on distributed RL (highlighted text in the introduction): We have added references to distributed RL post-training methods that could potentially improve the training computational cost of LOOP and PPO.
- GRPO comparison experiment (responding to reviewers yJgL and 3DSC): We conducted a small-scale experiment comparing GRPO with LOOP for the aesthetic task. Unfortunately, the base model used in our paper (Stable Diffusion 2.0) has been deprecated. We therefore ran experiments using an unofficial fork of SD 2.0, with results available at: https://anonymous.4open.science/r/TMLR-Rebuttal-4411 (see GRPO_vs_LOOP.png). Note that since this fork has undergone additional fine-tuning, the original results reported in the paper are not directly comparable, so we re-ran LOOP on the fork for consistency.
- Confidence intervals: We have added confidence intervals to all results reported on the test split (Tables 1, 2, and 3).
- Baseline comparison experiment (responding to reviewer J2GG): To compare the leave-one-out baseline proposed in LOOP with the running average-based baseline used in DDPO, we conducted a small-scale experiment (https://anonymous.4open.science/r/TMLR-Rebuttal-4411 see LOOP_RunAvg_vs_LOOP.png). As noted above, we re-ran LOOP on the SD 2.0 fork to ensure fair comparison.
- Pseudocode addition (responding to reviewer yJgL): We have added pseudocode for the LOOP training procedure in Appendix A.
- Enhanced experimental details: We have expanded the experimental details in Section 6.

We welcome continued discussion and are happy to address any additional questions or requests from the reviewers.

Thank you,

The Authors

---

### Decision · Action_Editor_YzeE · 2026-02-08

**Recommendation:** Accept with minor revision

**Additional Comments:**

The authors should ensure that their camera ready paper includes all points raised by the reviewers that were addressed in the rebuttal. Specifically, the plots provided in the rebuttal should be included in the paper.

Additionally:
- Adjust the wording in the discussion on page 2 regarding the efficiency-effectiveness trade-off in RLFT for diffusion. The second sentence refers to computational efficiency while the paper takes efficiency as sample efficiency as can be seen in a following paragraph. Please clarify this seemingly inconsistent use in the final version of the paper

- One reviewer raised concerns about the lack of KL penalty term which could lead to higher reward at the cost of sample diversity.  A suggested visualization is to show multiple generated samples per prompt to demonstrate diversity. The authors are encouraged to include this  in the paper or the appendix.

- LOOP is proposed as a novel method while an expert reader may see LOOP as an adaptation of existing methods in language models. This claim can be toned down although making this change is left to the discretion of the authors.

**Audience:**

Yes

**Audience Explanation:**

The paper proposes LOOP that is a method RL fine-tune  text-to-image (T2I) diffusion models. This work should be of interest to members in the TMLR community that work on diffusion and applying fine-tuning methods from language models to diffusion. This decision is supported by all three reviewers in their decision

**Claims And Evidence:**

Yes

**Claims Explanation:**

The paper proposes a method called Leave One Out PPO (LOOP) that is aimed at addressing the sample efficiency vs effectiveness tradeoff in reinforcement learning-based fine-tuning (RLFT) approach in text-to-image (T2I) diffusion. LOOP is built on the insight that REINFORCE, a simple method, requires variance reduction while PPO requires a careful choices of hyper parameters to work well. LOOP uses a leave-one-out approach for variance reduction that's adapted from prior RLOO while the method uses clipping from PPO to ensure the new policy is close to the old policy. The proposed method bears a strong resemblance to GRPO that the authors discuss in the paper. Empirical results on T2I-CompBench shows that the proposed method works better than the baseline DDPO method.

The reviewers agree that the proposed method is simple and that the empirical evidence provided in the paper support the effectiveness of LOOP. The main concerns raised during the reviews were that:
- The use of the term efficiency was not clear. This point which was addressed by the authors to mean sample efficiency in the rebuttal. But   there are a few sections in the introduction that need to be adjusted to ensure that efficiency does not mean compute efficiency.
- LOOP bears a strong resemblance to GRPO. The authors addressed this aspect by running an experiment during the rebuttal to show LOOP works better than GRPO in their application
- LOOP omits KL penalty term that may cause the fine-tuned model to collapse. The authors refer to prior work that suggests that KL penalty has a minimal effect on performance. While concerns remain, this does not detract from the strength of evidence provided in the paper to support the main claims.

We note that two of the three reviewers lean towards accepting the paper while the other reviews leans towards rejection. However, all three reviewers indicate that the claims made in the paper are supported by accurate, clear and convincing evidence. AE agrees with the reviewers  that the claims are supported by accurate, convincing and clear evidence while noting that there are a few items that the authors need to address in the final version of the paper.`

---

> ### Author Response · Authors · 2026-03-05
> **Camera ready version submitted**
>
> Dear AC and reviewers,
>
> We have updated the draft based on the area chair's comments. Specifically:
>
> - We have included all the points and plots discussed during the rebuttal.
>
> - We have modified the introduction to clarify that efficiency refers to sample efficiency throughout the paper.
>
> - We have added a qualitative visualisation to compare the diversity of different RL baseline methods as suggested by reviewer yJgL.
>
> We have uploaded the camera-ready version. We thank all reviewers and the AC for their reviews and the feedback on the draft.